

# Minute-scale power forecast of offshore wind turbines using single-Doppler long-range lidar measurements

Frauke Theuer[1], Marijn Floris van Dooren[1], Lueder von Bremen[2], and Martin Kühn[1]

[1]ForWind, Institute of Physics, University of Oldenburg, Küpkersweg 70, 26129 Oldenburg, Germany
[2]DLR Institute of Networked Energy Systems, Carl-von-Ossietzky-Straße 15, 26129 Oldenburg

**Correspondence:** Frauke Theuer (frauke.theuer@uni-oldenburg.de)

**Abstract.** Decreasing gate closure times on the electricity stock exchange market and the rising share of renewables in today's energy system cause an increasing demand for very short-term power forecasts. While the potential of dual-Doppler radar data for that purpose was recently shown, the utilisation of single-Doppler lidar measurements needs to be explored further to make remote sensing-based very short-term forecasts more feasible for offshore sites. The aim of this work was to develop a lidar-based forecasting methodology, which addresses a lidar's comparatively low scanning speed. We developed a lidar-based forecast methodology using horizontal plan position indicator (PPI) lidar scans. It comprises a filtering methodology to recover data at far ranges, a wind field reconstruction, a time synchronisation to account for time shifts within the lidar scans and a wind speed extrapolation to hub height. Applying the methodology to seven free-flow turbines in the offshore wind farm Global Tech I revealed the model's ability to outperform the benchmark persistence during unstable stratification, in terms of deterministic as well as probabilistic scores. The performance during stable and neutral situations was significantly lower, which we attribute mainly to errors in the extrapolation of wind speed to hub height.

## 1 Introduction

With the increasing penetration of renewable energies in the power system, the demand for very short-term power forecasts is continuously rising. Transmission System Operators (TSOs) need to ensure grid stability by balancing supply and demand of power at all times. In this regard, very short-term forecasts are an important tool to support power system management and reduce curtailment costs (Liang et al., 2016). Further, minute-scale forecasts hold significant value for energy market applications (Cali, 2011), especially with gate closure times nowadays being as short as only 5 minutes, for example in Germany, Belgium and France (EPEXSPOT, 2020). Also, the provision of ancillary services, e.g. the supply of reserve power by wind farms (50Hertz et al., 2016), would benefit from improved very short-term forecasts. Probabilistic forecasts additionally provide uncertainty information and are thus especially useful to support decision-making processes (Dowell and Pinson, 2016).

While for forecast horizons of several hours or days, typically physical models such as numerical weather prediction (NWP) models are used, on shorter time scales, i.e. lead times ranging from minutes to several hours, statistical models are applied (Giebel et al., 2011). For lead times of few to several hours, this includes mainly time series models, Kalman filters and Model Output Statistics (MOS) (Sweeney et al., 2019). The simplest statistical model for even shorter lead times is persistence, which





assumes the future value will be equal to the current one. Persistence is often referred to as a benchmark in very short-term forecasting (Würth et al., 2019). Other statistical models such as ARMA (autoregressive moving average) take a higher number of past values into account (Torres et al., 2005). ARIMA (autoregressive integrated moving average) models additionally consider past forecasting errors (Kavasseri and Seetharaman, 2009). Further, spatial correlation approaches, machine learning algorithms and neural networks are gaining importance for very short-term forecasts (Lenzi et al., 2018; Huang and Kuo,

2018). To overcome the limitations of individual models, combinations of different methodologies, so-called hybrid models, are being increasingly researched (Zhou et al., 2018).

As a promising alternative to statistical methods, recently very short-term forecasts based on remote sensing measurements have gained attention (Sweeney et al., 2019). The basic concept is to measure incoming wind fields in far distances upstream and thus several minutes before reaching the turbine or wind farm, allowing to derive wind speed and power forecasts on the

very short term. Lidar-based forecasts (LF) have for example been used by Valldecabres et al. (2018b) to predict nearshore wind speeds and they outperformed the benchmark persistence. Würth et al. (2018) used lidar measurements at an inland location to predict wind power, however, they were not able to outperform persistence, which the authors attributed to the complex terrain. A skilful probabilistic power forecast was recently developed by Valldecabres et al. (2018a), utilising dual-Doppler radar measurements performed by a radar system located at the shoreline, scanning the flow around an offshore wind

farm. Using the same radar set-up Valldecabres et al. (2020) moreover detected and probabilistically forecasted ramp events at free-stream as well as waked wind turbines.

While Doppler radars are capable of measuring in distances of up to 32 km (Nygaard and Newcombe, 2018) with high temporal and spatial resolution and provide volumetric 2D wind field information in case of a dual set-up (Hirth et al., 2017), such devices are also rather expensive, very large and thus not easily deployable at far offshore sites (Würth et al., 2019).

Studies also indicated their reduced data availability in comparison to lidars, especially during clear-air situations (Vignaroli et al., 2017; Hirth et al., 2017). Therefore, the use of Doppler lidar measurements instead of Doppler radar measurements is considered an interesting and probably more feasible alternative, especially with regard to offshore applications. Nowadays, compact industrial scanning lidar systems are able to measure at distances of up to 10 km (Leosphere, 2018). Hereby, the maximal measuring distance is closely related to the measurement accumulation time. To enlarge the maximal range of the

measurements, the accumulation time needs to be increased and thus the overall scanning speed is reduced. While radars can perform volumetric measurements, i. e. measurements with several different elevation angles, with a repetition time of the order of a few minutes, the slow scanning speed of current lidars restricts measurements to a single elevation angle when aiming to perform scans within a time frame of approximately one to two minutes. Consequently, depending on the positioning of the lidar system and the choice of elevation angle, the device's measuring height does not match the hub height of the turbine. Also,

platform or turbine movements can contribute to a static as well as dynamic misalignment (Bromm et al., 2018). Using such lidar measurements for wind speed and power prediction thus necessitates the use of a wind speed correction to hub height. As opposed to dual-Doppler measurements, the use of a single lidar device only allows retrieving one-dimensional wind speed information. Reasons for single-Doppler measurements are for example cost reduction or a wind farm layout that does not





favour a dual set-up. Consequently, a reliable wind speed reconstruction methodology is essential to retrieve horizontal wind
speed information from single lidar measurements.

Our objective in this paper is to investigate whether and how one can use long-range single-Doppler lidar measurements to
forecast the power of offshore wind turbines on short time horizons in a probabilistic manner. We adapt a remote sensing-based
forecast methodology to meet the requirements of single-Doppler lidar measurements. We especially implement adjustments
to account for i) low data availability in far ranges, ii) time shifts within the lidar scans and iii) deviations between measuring
height and hub height. We validate the method by means of a case study based on measurements at an offshore wind farm and
by distinguishing between different atmospheric conditions. To address their performance we compare lidar-based forecasts
against the benchmark persistence.

## 2 Planar long-range lidar measurements

For the purpose of forecasting, typically horizontal Plan Position Indicator (PPI) lidar scans, i. e. with an elevation angle of
$\varphi = 0°$, are used. Hereby, the lidar device can be placed either on the nacelle or transition piece (TP) of a wind turbine or a
nearby platform. The aim is to cover an area upstream of the wind farm, preferably in main wind direction. Scan parameters,
i. e. averaging time and azimuthal resolution are chosen to maximise the measurement distance while keeping the scanning time
as short as possible. Scan orientations need to be adjusted according to the wind direction. For each measurement, typically
the line-of-sight (LOS) velocity, carrier-to-noise-ratio (CNR) as well as azimuth angle, range gate, and time information are
available.

For the case study presented in Section 4 of this paper, such a typical set-up was used. Without loss of generality of the
methodology introduced in Section 3, we are describing the main parameters of this lidar campaign to provide a realistic
example. Lidar scans were performed at the offshore wind farm Global Tech I (GT I) located in the German North Sea from
August 2018 until February 2020 with a Leosphere Windcube 200S (Serial no. WLS200S-024) lidar system positioned on the
transition piece (TP) of the westerly located turbine T2 as depicted in Figure 1. The lidar was placed at a height of about $24.6\,\mathrm{m}$
above mean sea level. Scans were performed with an azimuthal resolution of $2°$, averaging time of $2\,\mathrm{s}$ per measurement, a pulse
length of $400\,\mathrm{ns}$ and range gates ranging from $500\,\mathrm{m}$ to $8000\,\mathrm{m}$ with $35\,\mathrm{m}$ spacing. The lidar scan spanned a sector of $150°$,
thus the duration of one scan was $T_\mathrm{tot} = 156\,\mathrm{s}$, i. e. measuring time $T_\vartheta = 150\,\mathrm{s}$ plus a measurement reset time of approximately
$T_\mathrm{r} = 6\,\mathrm{s}$. One of four different scan orientations (Figure 1 (b)) was chosen manually according to the wind direction. A more
detailed analysis of the lidar data will follow in Section 4.1.

Besides horizontal PPI lidar scans, high elevation scans with $\varphi = 13.57°$, measuring the inflow of turbine T2, were per-
formed. Here, it was measured with an azimuthal resolution of $1°$ and an averaging time of $0.2\,\mathrm{s}$ per measurement. At hub
height, measurements were performed with a distance to the rotor larger than $2.4D$ and therefore outside of the induction zone,
as recommended by the International Electrotechnical Commission's (IEC) standard for power curve measurements (IEC,
2017). A total azimuth range of $180°$, varying from $134°$ to $313°$ was spanned, which means it took about $36\,\mathrm{s}$ to perform one
scan and approximately $8\,\mathrm{s}$ to reset the measurement. Mean wind speeds and wind directions with an averaging period of $44\,\mathrm{s}$





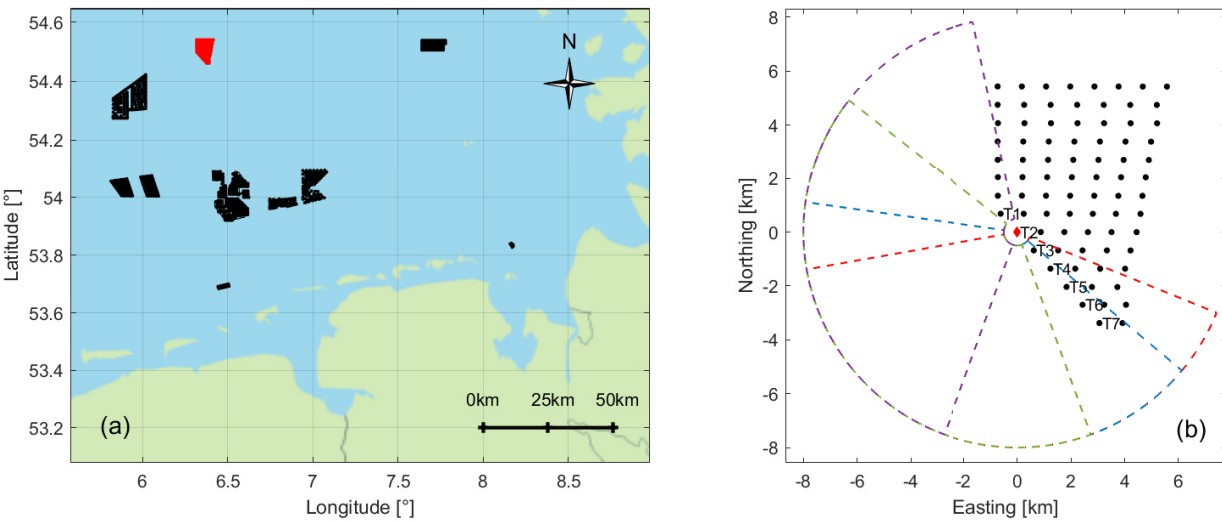

**Figure 1.** (a) Position of the offshore wind farm Global Tech I (GT I) in red. Other wind farms in the North Sea, which were operational during the measurement campaign, are shown in black. In (b) the layout of GT I is depicted with turbines marked as black dots. The lidar is positioned on the transition piece (TP) of turbine T2 marked in red. The four measuring trajectories are depicted in colour. Forecasts were generated for turbines T1-T7.

at hub height were determined by applying a Velocity Azimuth Display (VAD, see Section 3.1) algorithm to each scan. Only situations with wind directions ranging from 180° to 270° were considered for further analysis. The 44 s-mean wind speeds were used to construct a probabilistic power curve in Section 3.5.

## 95  3  Methodology

Figure 2 gives an overview of the proposed lidar-based forecast methodology. First, a wind field reconstruction algorithm was applied to retrieve horizontal wind field information from line-of-sight measurements of the angular scans (Section 3.1). To keep as much data from far ranges as possible, a dynamic data filtering approach was used. The low scanning speed required the time synchronisation within each lidar scan, which was realised by means of a propagation algorithm (Section 3.2). Following, 100 an advection technique was applied to determine a wind speed forecast (Section 3.3). The wind speed forecast was defined by selecting wind vectors arriving within a predefined area of influence (AoI). This set of wind vectors formed the basis of the probabilistic wind speed and power forecast. In the next step, wind vectors were extrapolated from measuring height to hub height (Section 3.4). Finally, hub height wind speeds were translated into a probabilistic power forecast utilising a probabilistic power curve (Section 3.5).





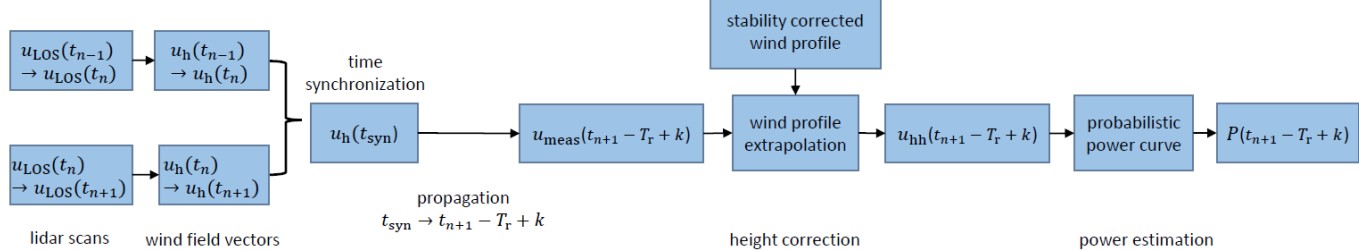

**Figure 2.** Schematics of the lidar-based forecast methodology. Line-of-sight wind speed measurements $u_{LOS,n}$ measured within a time interval $[t_n, t_{n+1} - T_r]$ were filtered and a wind field reconstruction was performed. Using two consecutive lidar scans, the horizontal wind speeds $u_{h,n}$ were then synchronised at time $t_{syn}$. A propagation technique was applied to propagate wind vectors to $t_{n+1} - T_r + k$ with end time of the scan $t_{n+1}$, measurement reset time $T_r$ and lead time $k$. Wind speed forecasts $u_{meas}$ were further extrapolated to hub height and transferred to power forecasts by means of a probabilistic power curve.

## 3.1 Lidar data filtering and wind field reconstruction

When performing lidar measurements several causes such as meteorological conditions, hard targets and device limitations can lead to invalid measurements. Typically, the carrier-to-noise-ratio (CNR) is used as an indicator for the backscattered signal's quality. Low CNR values hereby indicate low data quality and are commonly neglected by means of threshold filters (Aitken et al., 2012). However, when applying a CNR-threshold filtering approach, a significant amount of valid data, especially from far distances, is being excluded (Valldecabres et al., 2018b). As long measurement distances are most important for this work, we combined a CNR-threshold filter and a dynamic filtering approach. All measurements with $CNR > 0\,dB$ and $CNR < -30\,dB$ were neglected, measurements with $-26.5\,dB < CNR < -5\,dB$ were always considered valid and remaining values were filtered using the dynamic density filter developed by Beck and Kühn (2017). Here, CNR as well as line-of-sight (LOS) wind speed measurements were first normalised and sorted in a 2D plane before a 2D Gaussian function with standard deviations $\sigma_{CNR}$ and $\sigma_{LOS}$ and mean values $\mu_{CNR}$ and $\mu_{LOS}$ was fitted to the normalised values. Finally, those values positioned outside of an ellipse defined by the semi-axes $2.75\,\sigma_{CNR}$, $2.75\,\sigma_{LOS}$ and the centre position $\mu_{CNR}$ and $\mu_{LOS}$ were discarded.

After filtering, the global wind direction was determined by performing a VAD fit individually for each range gate in a certain scan. To do so, homogeneity across range gates was assumed and the vertical wind speed component neglected (Werner, 2005). Range gates with less than 15 valid lidar measurements were discarded. A one-dimensional wind speed projection on the prevailing wind direction of the range gate $r$ was performed using

$$u_h(r, \vartheta) = \frac{u_{LOS}(r, \vartheta)}{\cos(\vartheta - \chi(r))}. \tag{1}$$

Values with

$$75° < |\vartheta - \chi| < 105°, \tag{2}$$



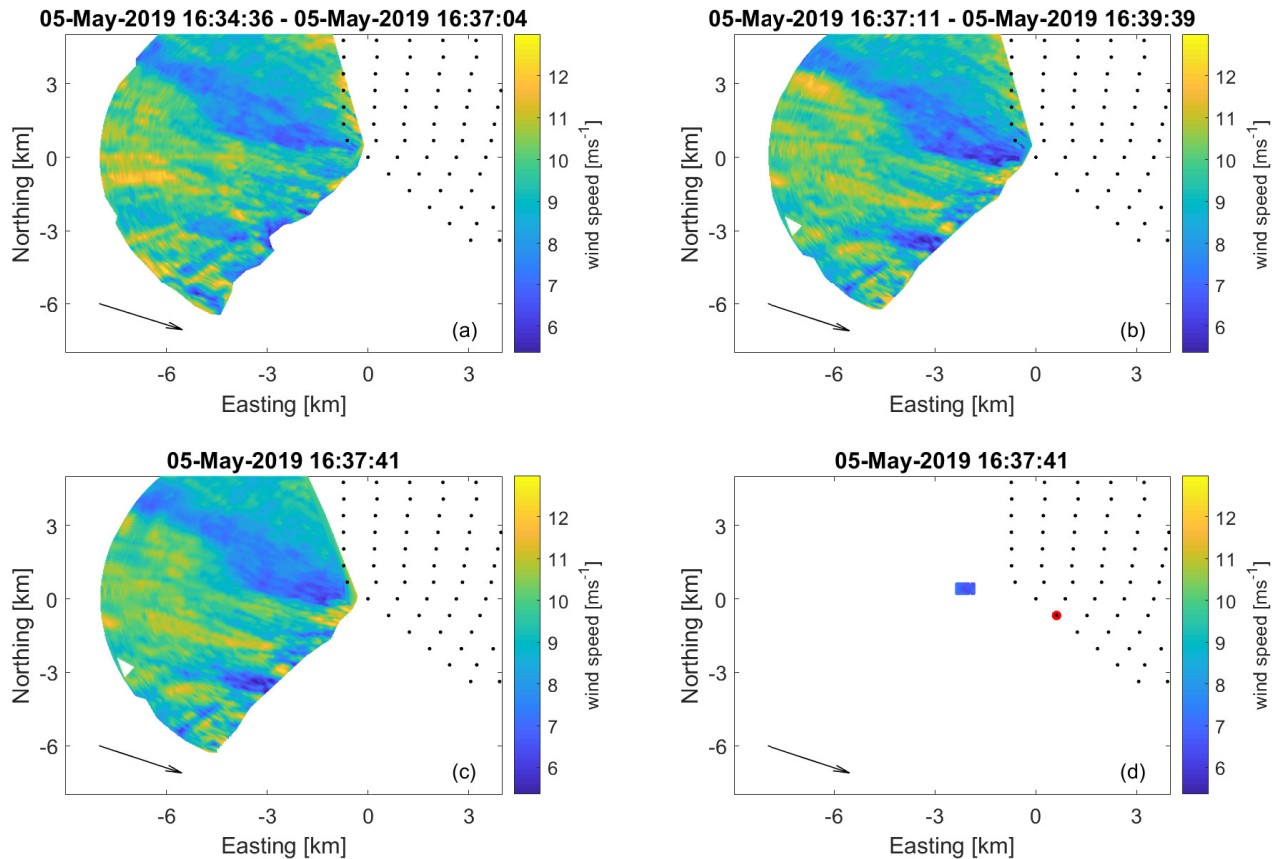

**Figure 3.** Time synchronisation and wind speed forecast of an exemplary scan. Subfigures (a) and (b) depict two consecutive lidar scans at GT I with black dots indicating the positions of the turbines. In (c) the time synchronised scan determined as a combination of the forward propagated scan (a) and the backwards propagated scan (b) is shown. In (d) a point cloud of wind vectors that will reach T3 marked in red after a forecasting horizon of 5 minutes±30 seconds is visualised. The mean wind direction of the scan is shown in black.

where $\vartheta$ denotes the azimuth angle of the lidar's scanner and $\chi$ the wind direction, were neglected as they show large error values due to the almost perpendicular orientation of wind direction and azimuth angle. We will refer to those as critical angles or critical region in the following. Apart from that, remaining outliers with values deviating more than 2.75 standard deviations $\sigma$ from the mean wind speed of the scan were neglected (Felder et al., 2018). Only scans with an overall data availability of at least 80 % were considered for the forecast. For further analysis, the results were interpolated onto a Cartesian grid with 25 m spacing. Figure 3 shows an example of a reconstructed wind field.





## 3.2 Time synchronisation of lidar scans

When the time shift within a lidar scan is larger than the averaging time (1 min) of the forecasted values, one cannot assume the scan to be quasi-instantaneous, which is commonly done when considering wind speed averages from lidar scans. Several approaches to account for the time shift within the scan have been tested, all aiming to synchronise the scan in time before applying the propagation methodology (Section 3.3). We found the most accurate results applying a time synchronisation developed by Beck and Kühn (2019), which is visualised in Figure 4. Here, lidar scans were propagated by means of a semi-Lagrangian advection technique. Propagated scans were generated with a temporal resolution of $\Delta T$. Each propagation was a combination of a forward- as well as a backwards-propagated scan, weighted according to a trigonometric function following the suggestion of Beck and Kühn (2019). The weighting was dependent on the time passed since the initialisation of the original scan. Hereby, backward propagations were only taken into account after one-fifth of the total scanning time $T_{\text{tot}}$. The total scanning time consists of the measuring time $T_\vartheta$ and the measurement reset time $T_{\text{r}}$. A 3D natural-neighbour interpolation (Sibson, 1981) was applied to the sequence of propagated scans, determining the horizontal wind speed $u_{\text{h}}$ across the scanned domain and at time $t_{\text{syn}}$. Figure 4 shows the current lidar scan initialised at time $t_n$ and the previous one initialised at $t_{n-1} = t_n - T_{\text{tot}}$ in blue. The scanning domain is visualised as azimuth angle $\vartheta$ over time. The propagation steps in between the two scans, performed with the temporal resolution $\Delta T$, are shown in grey. Backward and forward propagations are indicated as orange and green arrows. The red line indicates the synchronisation time step $t_{\text{syn}}$ at which the natural-neighbour interpolation was performed. For the purpose of forecasting, $t_{\text{syn}}$ should be chosen to stay within the region of the weighting function that puts no weight to backwards propagated scans, thus $t_{\text{syn}} \in [t_n, t_n + a\,\Delta T]$. Here $a$ denotes the maximal number of propagation steps possible, while avoiding backwards propagation. That means the time synchronisation can be performed by two consecutive lidar scans only, avoiding the need for a future scan. We chose the maximal $t_{\text{syn}} = t_n + a\,\Delta T$ to minimise the wind vector advection period as indicated by the black arrow in Figure 4.

## 3.3 Wind speed forecast

To generate a wind speed forecast the methodology developed by Valldecabres et al. (2018a) was utilised. A Lagrangian advection technique, based on the assumption that wind vectors propagate with their local horizontal wind speed and wind direction (Germann and Zawadzki, 2002), was applied. It was thus assumed that the wind field vectors do not change their trajectory with time. As a consequence of the wind field reconstruction explained in Section 3.1, the direction of wind vectors was the same for all azimuth angles and varied only with range gate. Apart from that, we neglected vorticity, mass conservation and diffusion (Germann and Zawadzki, 2002; Valldecabres et al., 2018a). To develop a wind speed forecast with lead time $k$, wind field vectors were propagated in time and space from their original position at the synchronised time step $t_{\text{syn}}$ to the last time step of the scan $t_{n+1} - T_{\text{r}}$ and further to $t_{n+1} - T_{\text{r}} + k$.

Vectors arriving within a previously defined area of influence around the turbine of interest and within a time interval of $t_{n+1} - T_{\text{r}} + k \pm 30\,\text{s}$ were selected and used for the wind speed forecast. An example of such a point cloud is shown in Figure 3 (d). The AoI was defined as a circle centred around the turbine's position and its radius was optimised by minimising the

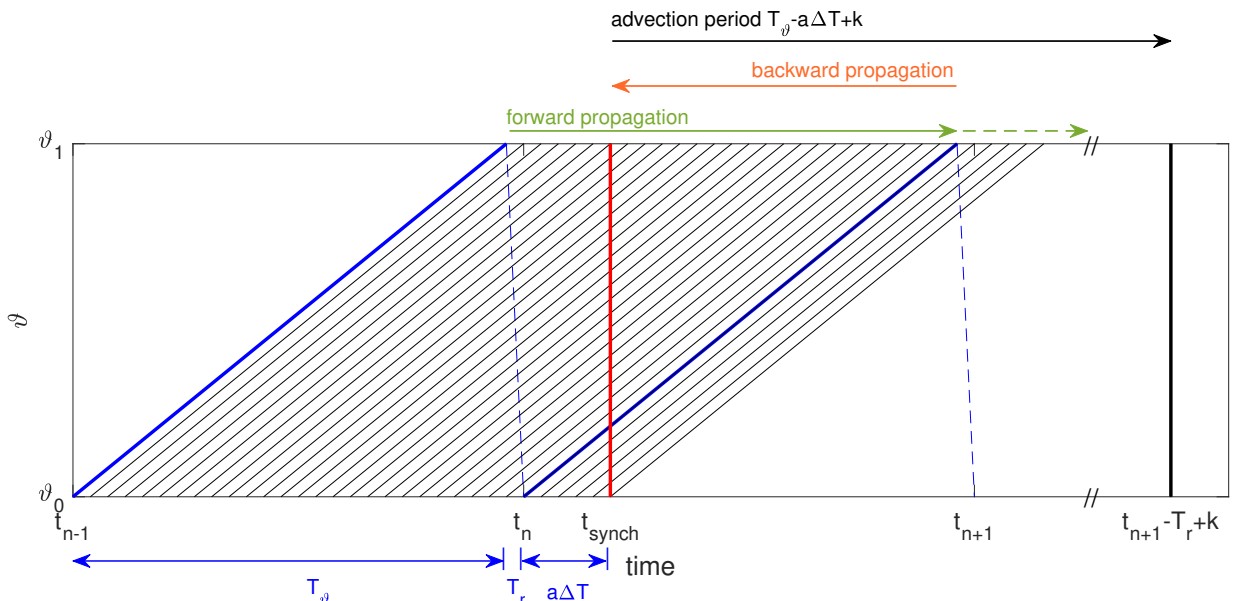

**Figure 4.** Time synchronisation of the lidar scan initialised at $t_n$ shown in blue to time $t_{\mathrm{syn}} = t_n + a\,\Delta T$ shown in red. The scanning domain is here visualised as azimuth angle $\vartheta$ over time. The synchronised scan is interpolated using propagated lidar scans with a temporal resolution of $\Delta T$. The measurement reset time $T_{\mathrm{r}}$ is indicated as dashed blue line. Propagated scans, which are shown in grey, are a combination of a forward and backward propagation weighted according to a trigonometric function. Green and orange arrows above the figure indicate to which of the propagated scans both forward and backward propagation and to which only a forward propagation contributes. The synchronised scan at $t_{\mathrm{syn}}$ can thus be reconstructed using only the two consecutive lidar scans initialised at $t_n$ and $t_{n-1}$. $t_{\mathrm{syn}}$ should be chosen to minimise the wind vector advection period to the forecast time $t_{n+1} - T_{\mathrm{r}} + k$, indicated as black arrow. Figure adapted from Beck and Kühn (2019).

average continuous ranked probability score (crps, see Section 4.4.1) (Gneiting et al., 2007) of a 1-minute-ahead wind speed forecast at a reference free-flow turbine as suggested by Valldecabres et al. (2018a). That means the forecast was optimised with respect to its probabilistic rather than its deterministic scores. Further, the minimum required amount of wind vectors reaching the turbine was determined by applying the same methodology. Forecasts that were based on less vectors are invalid.

    At this point, two orders of the methodological steps are possible, i. e. propagating wind vectors at varying heights that are
different from the height of interest to the target turbines before extrapolating to hub height or performing the extrapolation prior to the wind vector propagation. Each of the two possibilities is associated with specific errors. In this case study, we chose to propagate wind vectors before the wind speed extrapolation as this yielded more accurate results. The consequences of this approach will be discussed in Section 5.1.



### 3.4 Wind speed extrapolation to hub height

As the lidar was positioned at TP height, an extrapolation to the hub height is needed. A logarithmic wind profile including a stability correction $\Psi(\frac{z}{L})$ (Peña et al., 2008) was used to do so:

$$u_{\mathrm{h}} = \frac{u^*}{\kappa}\left(\ln\left(\frac{z}{z_0}\right) - \Psi\left(\frac{z}{L}\right)\right) \tag{3}$$

$$u^* = \sqrt{\frac{z_0\, g}{\alpha_{\mathrm{c}}}} \tag{4}$$

With the horizontal wind speed $u_{\mathrm{h}}$, roughness length $z_0$, height $z$, the gravitational acceleration $g$ and the Obukhov length $L$.

The friction velocity $u^*$ is expressed in terms of the Charnock parameter $\alpha_c$, which describes the relation between wind speed and roughness of the sea surface and was set to $\alpha_{\mathrm{c}} = 0.011$ as suggested by Smith (1980) for far offshore conditions. The Von Kármán-constant is defined as $\kappa = 0.4$.

The atmospheric stability for each lidar scan was determined using the methodology described by Sanz Rodrigo et al. (2019).

Air and sea surface temperature, pressure and relative humidity values were used to determine the virtual potential temperature difference $\Delta\Theta = \Theta_{\mathrm{TP}} - \Theta_0$ between TP height $\Theta_{\mathrm{TP}}$ and the sea surface $\Theta_0$ as well as the virtual temperature at sea level $T_{\mathrm{v}}$. The wind speed $u_{\mathrm{TP}}$ was defined as lidar measurements at the closest range gate of $500\,\mathrm{m}$. The stability estimation was performed using 30-minute moving averages of all variables. Here, first the Bulk Richardson number $Ri_{\mathrm{b}}$ was calculated, which was then transferred into the stability parameter $\zeta$ as defined by Grachev and Fairall (1997) and finally the Obukhov length $L$

according to Equations (5), (6) and (7).

$$Ri_{\mathrm{b}} = \frac{g}{T_{\mathrm{v}}}\frac{0.5\, z_{\mathrm{TP}}\Delta\Theta}{u_{\mathrm{TP}}^2} \tag{5}$$

$$\zeta = \begin{cases} \frac{10 Ri_{\mathrm{b}}}{1 - 5 Ri_{\mathrm{b}}} & Ri_{\mathrm{b}} > 0 \\ 10 Ri_{\mathrm{b}} & Ri_{\mathrm{b}} \leq 0 \end{cases} \tag{6}$$

$$L = \frac{0.5\, z_{\mathrm{TP}}}{\zeta} \tag{7}$$

For the calculation of the stability correction term $\Psi$ the definition of Dyer (1974) shown below was used.

$$\Psi = \begin{cases} 2\ln\left(\frac{1+x}{2}\right) + \ln\left(\frac{1+x^2}{2}\right) - 2\arctan(x) + \frac{\pi}{2} & L < 0, \text{ where } x = (1 - \gamma\frac{z}{L})^{1/4} \\ -\beta\frac{z}{L} & L \geq 0 \end{cases} \tag{8}$$

With $\beta = 6$ and $\gamma = 19.3$ as suggested by Högström (1988). The roughness length $z_0$ was determined by fitting the wind speed profile to the wind speed measurements $u_{\mathrm{TP}}$, using the calculated Obukhov length $L$.

With the height of the measurement $z_{\mathrm{meas}}$ the wind speed at hub height $u_{\mathrm{hh}}$ can then be expressed as:

$$u_{\mathrm{hh}} = u_{\mathrm{meas}}\frac{\ln\left(\frac{z_{\mathrm{hh}}}{z_0}\right) - \Psi\left(\frac{z_{\mathrm{hh}}}{L}\right)}{\ln\left(\frac{z_{\mathrm{meas}}}{z_0}\right) - \Psi\left(\frac{z_{\mathrm{meas}}}{L}\right)} = u_{\mathrm{meas}}\, c_{\mathrm{h}} \tag{9}$$



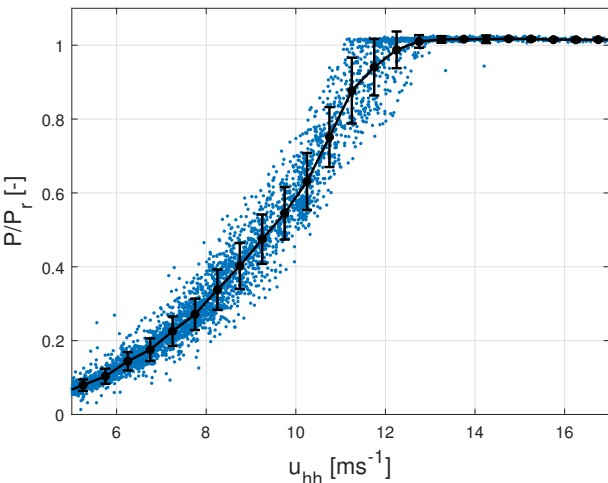

**Figure 5.** Normalised probabilistic power curve of wind turbine T2 with average power and its standard deviation in black for each wind speed interval.

In the following, we will refer to $c_h$ as the height extrapolation factor.

### 3.5 Probabilistic wind power forecast

The forecasted wind speed distribution was finally transformed into a wind power distribution. To do so, a probabilistic power
curve constructed using high elevation lidar scans (Section 2) and high-frequency SCADA power data of turbine T2 (Section 4.1) was applied. Usually, 10-minute wind speed and power averages are used to construct power curves, however, we used 44-second-mean values, in accordance with the measurement time per scan, to capture the power curve's associated uncertainties more accurately (Gonzalez et al., 2017). Wind speed values were air density corrected as described by Ulazia et al. (2019) and according to IEC 61400-12-1 (IEC, 2017). Air pressure and temperature values were hereby corrected to hub height applying
temperature gradients of the ISO standard atmosphere as suggested by ISO2533 (ISO, 1987). The mean value and standard deviation of power within wind speed intervals of $0.5\,\mathrm{m\,s^{-1}}$ width were determined (Gonzalez et al., 2017). These values were further used to define a normal cumulative distribution function (cdf) of power for each wind speed interval. Figure 5 shows the normalised probabilistic power curve with standard deviations of power indicated by error bars. For each value of the forecasted wind speed distribution, i. e. for each wind vector reaching the area of influence, one power value was randomly
selected using the normal cdf of its corresponding wind speed interval. A resampling technique with replacement (Efron, 1979) was applied to the resulting power distribution, randomly selecting $10\,000$ power values, as suggested by Valldecabres et al. (2018a).





## 4   Results

In the following, we will first introduce the case study at the offshore wind farm Global Tech I, then analyse the method's ad-
vantages and limitations, and afterwards assess the quality of a 5-minute-ahead lidar-based deterministic as well as probabilistic
wind power forecast of the free flow turbines T1-T7, based on the mentioned case study.

### 4.1   Case study at the offshore wind farm GT I

Power forecasts at Global Tech I were analysed as a case study. The wind farm consists of 80 wind turbines of type Adwen
5-116 with a rotor diameter of $D = 116\,\mathrm{m}$, a hub height of $z_{\mathrm{hh}} = 92\,\mathrm{m}$ and a rated power of $P_{\mathrm{r}} = 5\,\mathrm{MW}$. The total capacity of
the wind farm is $P_{\mathrm{total}} = 400\,\mathrm{MW}$. The 1 Hz SCADA data, including power and wind direction values of all wind turbines, as
well as information regarding the turbines' operational status, was available for the period of the measurement campaign. Wind
speed values were not measured but estimated by the SCADA system based on power, pitch angle and the turbine power curve.
Further, information regarding the SCADA data quality was available and used to remove low-quality data. In the following
analysis, we used 1-minute-mean values of wind speed and power within the interval $t \pm 30\,\mathrm{s}$ to validate wind speed as well as
power forecasts for seven wind turbines in the first south-westerly row marked in Figure 1. We refer to those turbines as T1-T7
in the following.

A forecast was generated for each lidar scan, thus with a temporal resolution of approximately 2.5 minutes. Forecasts within
the interval 08.03.2019 to 31.05.2019 were evaluated. Here, we only considered situations in power production mode below
rated wind speed. For further analysis, only scans with a total spatial availability of at least 80 % after applying the filtering
algorithms (Section 3.1) were considered. The total availability is considered 100 % if data at all measured range gates and
azimuth angles between $140°$ and $300°$ is valid. Missing data beyond these azimuth limits was considered not to impact the
quality of the forecast gravely and thus neglected when determining the total spatial availability. In total, 17 024 lidar scans
with a mean availability of 89.7 % were used for the analysis. The wind speed and direction distribution of those situations
considered are visualised in Figure 6. North-westerly winds from $250°$ to $320°$ were identified as the prevailing wind direction.
Wind speeds mainly lay between $6\,\mathrm{m\,s^{-1}}$ and $12\,\mathrm{m\,s^{-1}}$. As a consequence of the wind farm's layout, we only used scans with
wind directions $130° < \chi \leq 350°$, indicated as grey shaded area in Figure 6.

To perform the time synchronisation an interpolation time step $\Delta T = 6\,\mathrm{s}$ was chosen. With a scanning time of $T_{\mathrm{tot}} = 156\,\mathrm{s}$,
we chose the synchronisation time as $t_{\mathrm{syn}} = t_n + 5\Delta T = t_n + 30\,\mathrm{s}$ in order to avoid the need for a backwards propagation as
explained in section 3.2. Time synchronised wind vectors were propagated with a lead time of $k = 300\,\mathrm{s}$ to generate a wind
speed forecast. For a forecast to be valid, at least a number of $Z = 20$ wind vectors needed to be available. The radius of the
area of influence was set to $R_{\mathrm{AoI}} = 0.2D = 23.2\,\mathrm{m}$, following the methodology described in Section 3.3, with T2 as reference
turbine.

$L$ was determined using meteorological measurements: Air pressure, humidity and air temperature measurements were per-
formed using two sensors (Vaisala PTB330 and Vaisala HMP155 respectively) from July 2018 until February 2020, both

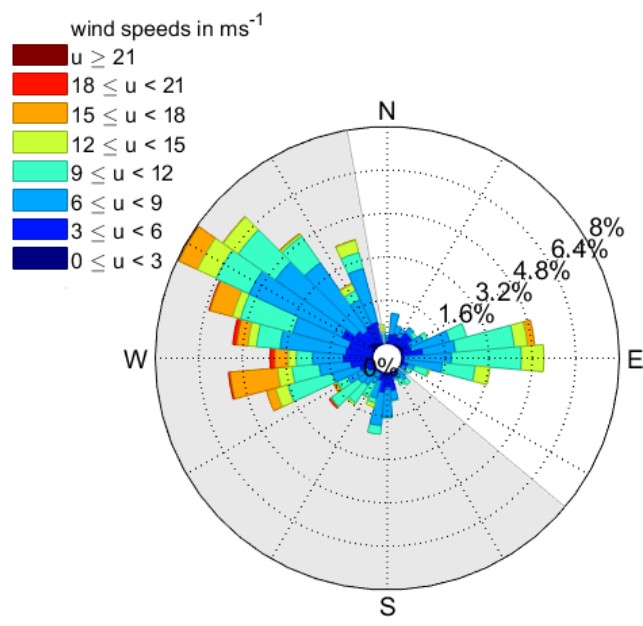

**Figure 6.** Wind speed and direction distribution of the used data set at the wind farm Global Tech I. Shown are mean wind speed and direction values for all lidar scans with a data availability of at least $80\,\%$ within the period from 08/03/2019 to 31/05/2019. The grey shaded area indicates wind directions that were considered for further analysis.

positioned at the height of the lidar at about $24.6\,\mathrm{m}$. Additionally, sea surface temperature (SST) data, which showed a good agreement with on-site buoy measurements performed at an earlier time (Schneemann et al., 2020), was available from the OSTIA data set (Good et al., 2020). SST data is available at noon every day and was linearly interpolated to match the timestamps of the lidar scans. $L$ was then used to extrapolate wind vectors from measuring height to hub height following Section 3.4.

During the measurement campaign, a slight elevation misalignment of the lidar was detected. Using a so-called sea surface levelling method the magnitude of pitch and roll of the lidar, i. e the tilt of the geographical coordinate system, was determined as proposed by Rott et al. (2017). The inclinations were hereby found to be related to the mean wind speed and wind direction, i. e. the thrust respectively the yaw orientation of the turbine. Pitch and roll, defined as clockwise rotations around the x- respectively y-axis, were $0.02°$ and $-0.11°$ for the turbine in idling mode and $0.02°\pm0.15°$ respectively $-0.11°\pm0.11°$ during

power production, depending on the mean wind speed. As even small errors in the elevation will lead to large differences in the measurement height, especially for far measurement distances, we accounted for the misalignment by means of a correction function. The correction function used the power production of the turbine and the mean wind direction to determine pitch and roll. These values were then used to estimate the corrected measuring height across the scanned area. Height differences due to the curvature of the Earth were considered as well. An additional uncertainty was introduced by the tide, which varied

approximately $\pm0.6\,\mathrm{m}$. For simplicity, we neglected this influence.

    The measuring height $z_{\mathrm{meas}}$ in Equation (9) therefore varied with range gate and azimuth for each scan. Heights of wind vectors





contributing to wind speed forecasts in this analysis spanned between a height of 12 m and 65 m, with a mean height of 36 m. Wind vectors extrapolated to hub height were in a final step transformed to wind power values using the methodology and power curve introduced in section 3.5. For the evaluation of the probabilistic wind power forecasts, we distinguished between stable

or neutral and unstable atmospheric stratification. Situations with values of $-1\,000\,\text{m} < L < 0\,\text{m}$ were classified as unstable, while those with $0\,\text{m} < L < 1\,000\,\text{m}$ were defined as stable (Van Wijk et al., 1990). All other cases were defined as neutral.

### 4.2  Evaluation of Methodology

Here, we aim to present the results of the individual methodical steps introduced previously. We assessed how the use of single-Doppler measurements and the low scanning speed affected the lead time, availability and skill of the forecast. Further,

the impact of the extrapolation to hub height was analysed (Theuer et al., 2020). Finally, we examined how the single-Doppler data may have influenced the prediction intervals of the forecast.

The data availability of all valid lidar scans dependent on range gate, applying different filtering methodologies, is compared in Figure 7. Clearly, the availability of data was increased for far ranges when applying the density filter (red line) as compared to a CNR-threshold filter (blue line) with $-26.5\,\text{dB} < \text{CNR} < -5\,\text{dB}$. While the data availability at a range gate of 7 km has

already decreased to 42 % for the threshold filter, it still lies at 73 % when using the density filter. Also at very close range gates from 500 m to 1450 m, the availability was increased from about 95 % to almost 100 %. The green line depicts the data availability after applying the density filter and additionally neglecting all other invalid data. That included the removal of wind speed outliers, however, the dominant effect was the omission of values within the critical region as described in Section 3.1. For the given measurement set-up and range gates up to 6 km, the availability was reduced to approximately 85 % of the

density filtered data. At 7 km it has decreased to 61 %. As the data availability was already reduced for far distances, the impact of further filtering was smaller compared to near ranges with higher data availability.

The number of observations at each measurement point in the polar coordinate system of the lidar before filtering is shown in Figure 8 (a). Clear differences in the number of observations are visible as a consequence of the four scanning trajectories of the lidar (Figure 1 (b)). In accordance with the wind direction distribution (Figure 6), north-westerly sectors were covered

more frequently than southerly sectors. Figure 8 (b) visualises the number of observations after filtering not only dependent on range gate, but also on the azimuth angle. The single-Doppler set-up caused the need to apply a VAD-fit and as a consequence to filter certain regions of the scan, earlier referred to as critical regions. As an effect, data availability is significantly reduced. For instance, for the vicinity of the prevailing wind direction of approximately $300°$ (Figure 6) the critical region defined by Equation 2 ranges from $15°$ to $45°$ and $195°$ to $225°$, i. e. the sectors perpendicular to the wind direction. Consequently, at

an azimuth of $220°$ the availability was degraded from $1\,085\,000$ observations to only $732\,200$. Figure 9 (a) shows how the availability of measurements impacted the number of valid forecasts for the turbines T1-T7, depending on the scan's mean wind direction. The black line indicates the total number of valid scans, thus the maximal number of possible forecasts, available for the wind direction intervals. Compared to this, the number of valid forecasts is much lower for wind directions larger than $250°$ for turbines T5, T6 and T7. The forecast's quality is also worse for those wind directions, especially for T6 and T7 (Figure 9

(b)). Here, the lidar scans mainly covered the north-westerly inflow direction of the wind farm. Consequently and due to the

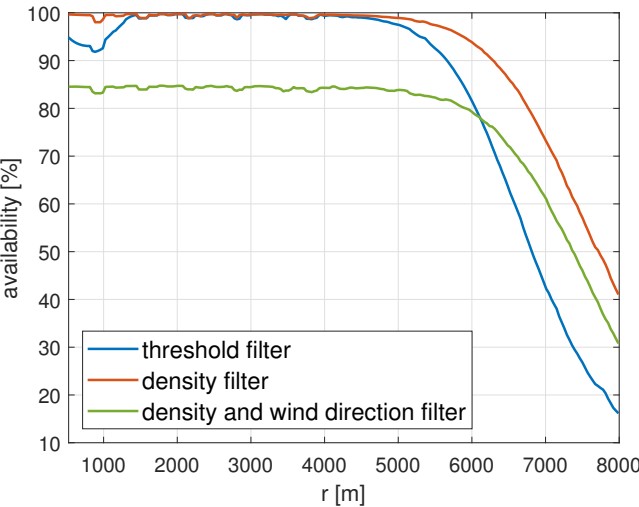

**Figure 7.** Data availability dependent on range gate when applying the threshold filter (blue) and the density filter (red). In green the availability after applying the density filter and neglecting the critical angles and wind speed outliers is depicted.

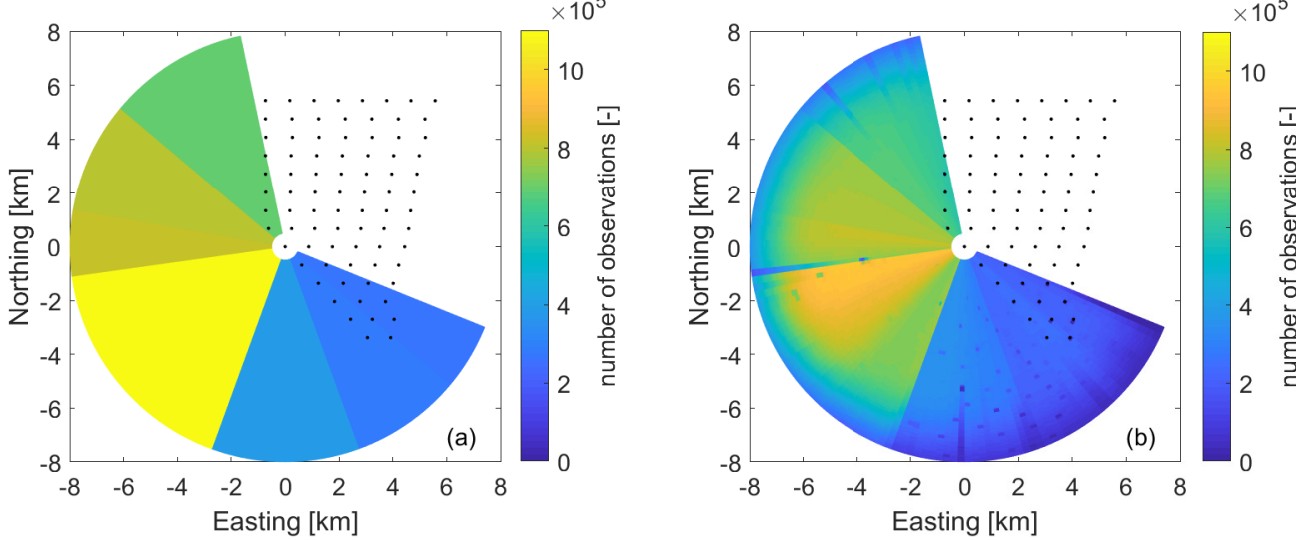

**Figure 8.** Distribution of measurements across the measurement domain as a result of varying lidar trajectories. The number of observations at each measurement point in the polar coordinate system of the lidar is visualised (a) before and (b) after data filtering. Filtering includes the density filter, the exclusion of critical regions and filtering wind speed outliers.

layout of the wind farm, especially the area from which wind vectors are propagated to turbines T5, T6 and T7 was not covered well by the lidar scan, resulting in reduced forecast availability and quality.

Also for wind directions ranging from 160° to 200°, the quality of forecasts at T5, T6 and T7 is lower compared to the other turbines. A similar problem occurred here as the turbines are placed within the scan area (Figure 8). For the mentioned wind





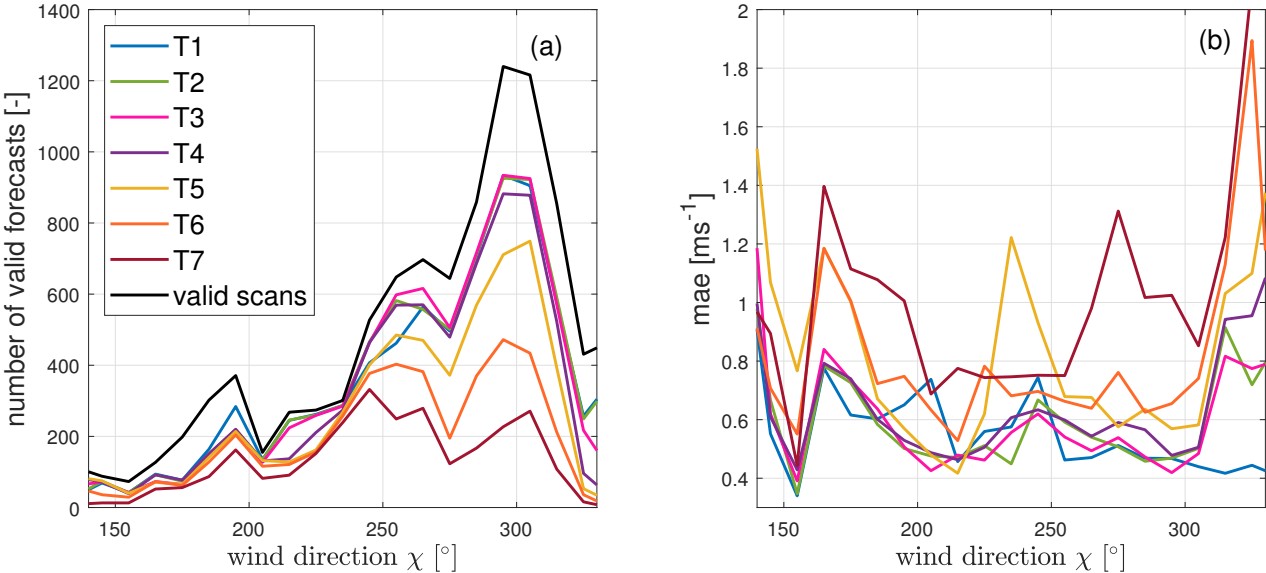

**Figure 9.** Number of valid forecasts (a) and mae (b) for turbines T1-T7 as a function of wind direction $\chi$ sector of $10°$ width. The black line in (a) depicts the maximal number of possible forecast, i. e. the number of valid scans, available for each wind direction.

directions the area from which wind vectors can be propagated is thus considerably smaller, resulting in less available vectors and consequently higher forecast errors. Furthermore, vectors contributing to the wind speed forecast originate from far range gates. Typically, in higher range gates the lidar has larger measurements errors and additionally the tangential distance between measurements is greater, resulting in less accurate interpolations to the Cartesian grid.

The application of the time synchronisation method introduced in Section 3.2 extended the wind vector propagation time of the 5-minute-ahead forecast from $300\,\mathrm{s}$ to $420\,\mathrm{s}$. The synchronisation time $t_{\mathrm{syn}}$ was set to $30\,\mathrm{s}$ after the initialisation of the scan (Section 4.1). That means, to reach the last time step of the scan at $150\,\mathrm{s}$, not considering the measurement reset time, a propagation of additional $120\,\mathrm{s}$ was required. The total scanning time hereby determines the additional propagation time. That means low scanning speeds reduce the maximal forecast lead times, in this case by 2 minutes.

Based on the data availability at different range gates, the maximal possible lead time of the forecast was determined. Even though we only considered partial load situations, i. e. situations with mean wind speeds up to $12\,\mathrm{m\,s^{-1}}$, higher wind speeds may have contributed to the wind speed distribution. Excluding those by choosing a too large lead time would thus falsify the results. Considering $15\,\mathrm{m\,s^{-1}}$ wind speeds, the measuring range needs to extend to $4\,500\,\mathrm{m}$ for 5-minute-ahead forecasts and to $9\,000\,\mathrm{m}$ for predictions with a lead time of 10 minutes. Taking into account the additional propagation time of $120\,\mathrm{s}$, the

measuring range would even need to be extended to $6\,300\,\mathrm{m}$ and $10\,800\,\mathrm{m}$ respectively. Due to the layout of the wind farm, turbines are often placed inside the scan, which can – depending on wind direction – lead to a reduction of the maximal possible advection distance. Taking this into account, combined with the fact that the data availability decreases with range, it is not

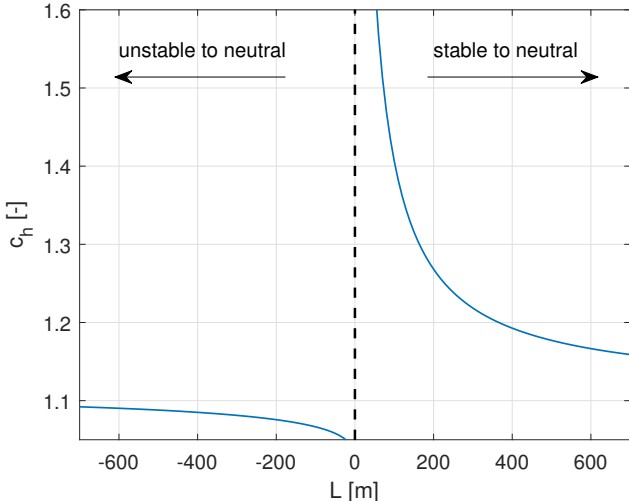

**Figure 10.** Dependence of the height extrapolation factor $c_{\mathrm{h}}$ on the Obukhov length $L$. In this example an extrapolation from 24.6 m to 92 m with a roughness length of $z_0 = 0.0002$ m is shown.

possible to generate forecasts with lead times larger than 5 minutes using the available lidar scans.

The wind speed extrapolation to hub height is, following the method introduced in Section 3.4, mainly dependent on stability. Figure 10 shows the dependence of the height extrapolation factor $c_{\mathrm{h}}$, calculated with Equation 9, on Obukhov lengths $L$ assuming an extrapolation from a height of 24.6 m to 92 m and a roughness length of $z_0 = 0.0002$ m. While the slope of the curve becomes very small when approaching Obukhov lengths $L$ with large magnitudes, thus neutral situations, especially for very stable cases $L \rightarrow 0$ the change of the correction factor with $L$ is very large. This consequently means misestimations of

Obukhov length $L$ have a larger impact on the wind speed extrapolation in stable situations. In order to determine this effect, we distinguished between stability cases in the following analysis. While during 55.5 % of the valid scans unstable atmospheric stratification was observed, in 18.2 % the atmosphere was defined as neutral. Stable situations were observed in 26.3 % of the cases. To be able to evaluate unstable cases, during which we expect the highest errors in persistence as compared to stable and neutral ones, separately and to keep the number of analysed cases similar we chose to combine stable and neutral situations for

the analysis.

## 4.3    Deterministic wind power forecast

### 4.3.1    Deterministic forecast evaluation

Wind speed point forecasts were calculated as the mean of the predicted wind speed distributions. Forecasts (fc) were verified with 1-minute-mean SCADA data (obs) and using the root mean squared error (rmse), mean absolute error (mae) and bias. $N$





denotes the total number of forecasts considered.

$$\text{rmse} = \sqrt{\frac{1}{N} \sum_{i=1}^{N} (\text{fc}_i - \text{obs}_i)^2} \tag{10}$$

$$\text{mae} = \frac{1}{N} \sum_{i=1}^{N} |(\text{fc}_i - \text{obs}_i)| \tag{11}$$

$$\text{bias} = \frac{1}{N} \sum_{i=1}^{N} (\text{fc}_i - \text{obs}_i) \tag{12}$$

As a reference, the benchmark persistence was used, which assumes the future value at $t + k$ equals the current value at time
$t$, i. e. $\text{fc}(t + k) = \text{obs}(t)$.

### 4.3.2   Unstable stratification

Figure 11 compares 1-minute-mean SCADA wind power values of turbine T3 with persistence and the lidar-based forecasts
(LF) in unstable atmospheric conditions. Both methods show an overall good agreement between forecast and observation with
$R^2 = 0.80$ respectively $R^2 = 0.86$, with the LF's scatter being slightly smaller than that of persistence. The LF outperforms
persistence in terms of rmse and mae. The lidar forecast's bias of $0.52\,\%$ is slightly larger than that of persistence with $0.31\,\%$.
The magnitude of the error is increasing with increasing power for both persistence as well as the lidar-based forecast. As the
wind speed forecasting error was not found to increase with wind speed, the increase of error with power is attributed solely to
the cubic nature of the power curve.

Table 1 summarises the results of turbines T1-T7 for unstable situations for all valid forecasts and also shows the scores
for only simultaneously available forecasts. During unstable atmospheric stratification and for all available forecasts, the LF
outperforms persistence for turbines T1-T4 in terms of rmse and mae, with the lowest rmse observed for T1 and the largest
improvement as compared to persistence for T3 with $20.1\,\%$. The bias of those turbines is slightly larger than of persistence but
rather small and not suggesting a systematic over- or underestimation of power caused by the model. T5 shows lower forecast
skill and outperforms persistence only in terms of rmse. The quality of the LF at T6 and T7 is below that of T1-T4 with a
strongly reduced number of valid forecasts $N$. We attribute this to the turbines' position in an area not covered well by the lidar
scans, which means fewer wind vectors can be propagated to the target turbines (Section 4.2). In Figure 8 (b) it can be observed
that especially regions very close to these turbines have low data availability. Low wind speeds, possibly originating from those
areas, are thus not represented well in the wind speed distributions causing an overestimation of wind speed and power. Also
for only simultaneously available forecasts, the LF outperforms persistence for all turbines except T6 and T7. The difference
in quality is less distinct in that case, with the rmse increasing by a factor of 1.8 instead of 2.4 from T3 to T7. While therefore,
some of the quality differences observed for all available cases can be explained by the varying time intervals considered, this
also confirms that forecast accuracy depends on the availability of wind vectors.





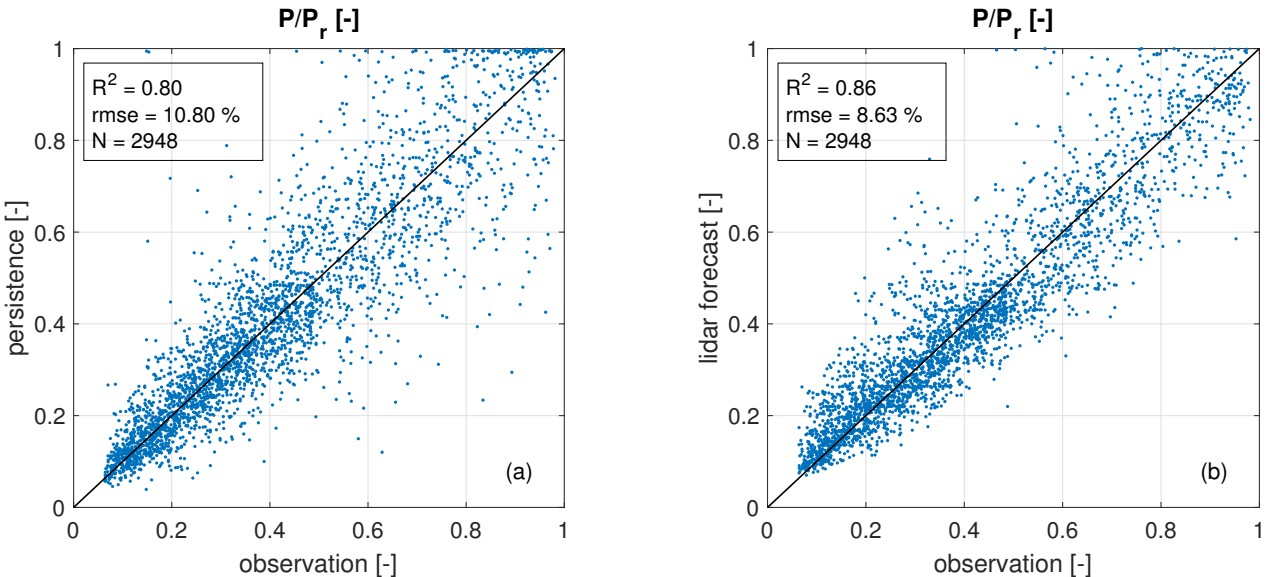

**Figure 11.** Comparison of 5-minute-ahead power forecasts at turbine T3 with 1-minute-mean SCADA data for (a) persistence and (b) the lidar-based forecast for **unstable** stratification. Values are given in % of the turbine's nominal power

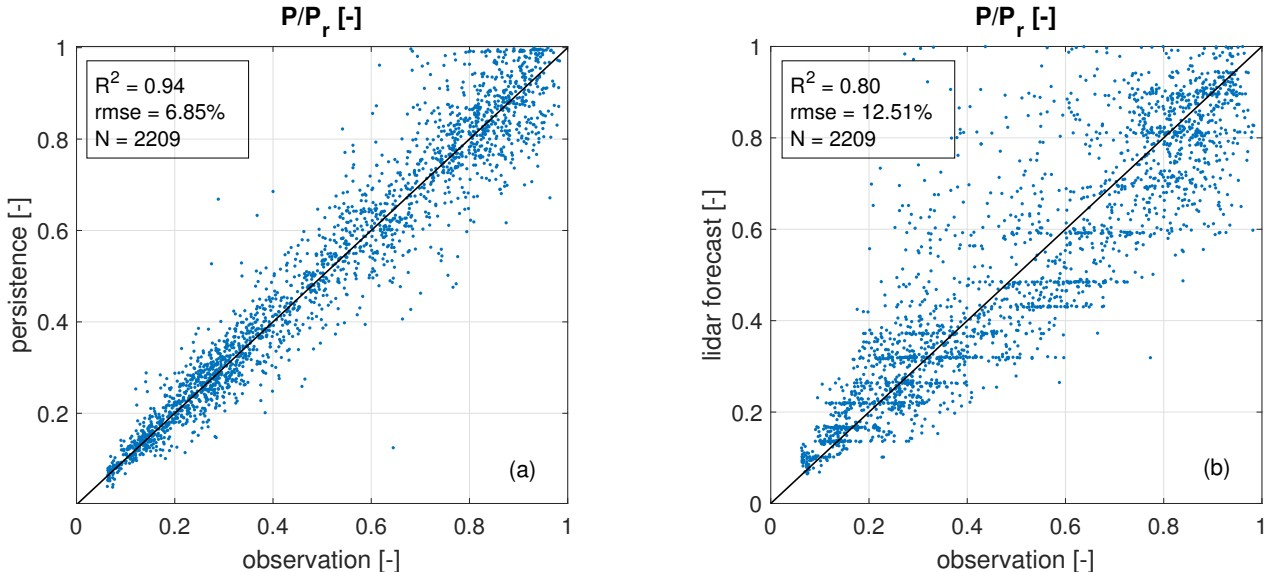

**Figure 12.** Comparison of 5-minute-ahead power forecasts at turbine T3 with 1-minute-mean SCADA data for (a) persistence and (b) the lidar-based forecast for **stable and neutral** stratification. Values are given in % of the turbine's nominal power





**Table 1.** Number of valid forecasts $N$, rmse, mae and bias for turbines T1-T7 for the lidar-based forecast and persistence during **unstable** stratification for all available forecasts and all simultaneously available ones. Scores are given in % of the turbines' nominal power with lowest values in bold.

|  |  | T1 | T2 | T3 | T4 | T5 | T6 | T7 |
|---|---|---|---|---|---|---|---|---|
|  | N | 2510 | 2840 | 2948 | 2710 | 2048 | 640 | 297 |
| rmse [%] | lidar-based | **7.20** | **8.81** | **8.63** | **9.54** | **11.39** | 13.87 | 21.07 |
|  | persistence | 8.37 | 9.99 | 10.80 | 10.95 | 12.09 | **10.94** | **11.35** |
| mae [%] | lidar-based | **5.02** | **6.20** | **6.17** | **6.91** | 8.53 | 10.53 | 16.67 |
|  | persistence | 5.45 | 6.75 | 7.34 | 7.72 | **8.63** | **8.28** | **8.88** |
| bias [%] | lidar-based | -1.16 | 0.86 | 0.52 | -0.24 | -1.65 | 4.81 | 10.37 |
|  | persistence | **0.41** | **0.36** | **0.31** | **0.26** | **0.09** | **-1.54** | **-1.69** |
|  | N | 45 | 45 | 45 | 45 | 45 | 45 | 45 |
| rmse [%] | lidar-based | **9.26** | **9.75** | **7.91** | **7.73** | **8.23** | 10.54 | 14.08 |
|  | persistence | 10.52 | 9.80 | 12.78 | 10.14 | 12.10 | **9.94** | **12.28** |
| mae [%] | lidar-based | **6.77** | **6.84** | **5.92** | **6.28** | **5.97** | 7.63 | 11.00 |
|  | persistence | 8.08 | 6.97 | 9.72 | 8.10 | 10.08 | 7.92 | **9.32** |
| bias [%] | lidar-based | -4.31 | -1.20 | **-2.69** | -1.01 | -2.37 | **-0.29** | -2.45 |
|  | persistence | **-1.78** | **0.10** | -3.88 | **-0.31** | **-2.28** | -2.08 | -3.75 |

**Table 2.** Number of valid scans $N$, rmse, mae and bias for turbines T1-T7 for the lidar-based forecast and persistence during **stable and neutral** stratification. Scores are given in % of the turbines' nominal power with lowest values in bold.

|  |  | T1 | T2 | T3 | T4 | T5 | T6 | T7 |
|---|---|---|---|---|---|---|---|---|
|  | N | 2013 | 2197 | 2209 | 2102 | 1273 | 1219 | 554 |
| rmse [%] | lidar-based | 12.57 | 12.23 | 12.51 | 13.16 | 13.52 | 14.61 | 16.01 |
|  | persistence | **6.87** | **6.90** | **6.85** | **7.29** | **7.78** | **6.56** | **6.36** |
| mae [%] | lidar-based | 8.77 | 8.64 | 8.95 | 9.46 | 9.72 | 10.77 | 11.88 |
|  | persistence | **4.56** | **4.80** | **4.75** | **5.08** | **5.14** | **4.64** | **4.26** |
| bias [%] | lidar-based | 1.26 | 1.39 | **0.18** | **0.01** | 1.05 | 2.85 | 6.66 |
|  | persistence | **0.56** | **0.55** | 0.39 | 0.78 | **0.55** | **0.44** | **0.35** |

### 4.3.3 Stable and neutral stratification

Figure 12 shows the comparison of SCADA data and LF as well as persistence forecasts for stable and neutral atmospheric

conditions at turbine T3. While the overall agreement between observation and forecast is very good for persistence, larger

scatter, a higher rmse and mae are observed for the LF. Generally, persistence clearly outperforms the LF during stable and



neutral conditions in terms of rmse and mae as summarised in Table 2. While the lidar forecast's bias for T3-T5 is lower than that of persistence, it shows a large overestimation of power, especially for T6 and T7. Similar to unstable cases, the quality and number of valid lidar-based forecasts decreases for turbines positioned in areas not well covered by the lidar scan. As especially areas close to the turbines are not represented well (Figure 8 (b)), wind speed and power are being overestimated.

The quality of persistence is much better compared to unstable situations, due to lower wind speed fluctuations characteristic in stable situations (Stull, 2017). The lidar forecast's skill, however, is considerably lower compared to unstable situations. We attribute this to the extrapolation of wind speed to hub height. Variations in Obukhov length $L$ and measuring height $z_{\mathrm{meas}}$ have a larger impact on the height extrapolation factor for stable situations compared to unstable situations, leading to larger errors in case of misestimations. We will discuss this in more detail in section 5.1.

### 4.4 Probabilistic wind power forecast

#### 4.4.1 Probabilistic forecast evaluation

Probabilistic forecasts are generally evaluated by means of their sharpness and calibration. Sharpness describes the broadness of its distribution, while calibration estimates the consistency between the statistics of forecasts and observations (Gneiting et al., 2007). Both calibration and sharpness are estimated with the average crps:

$$\overline{\mathrm{crps}} = \frac{1}{N} \sum_{i=1}^{N} \int_{-\infty}^{\infty} [F_i(x) - H(x - x_{0,i})]^2 dx \tag{13}$$

Here, $F$ denotes the cdf of the forecasted wind power, $x_0$ the observed wind power and $H$ the Heaviside step function with $H(x - x_0) = 0$ for $x < x_0$ and $H(x - x_0) = 1$ otherwise.

To assess the forecast's calibration quantile-quantile reliability diagrams (Hamill, 1997) were used. A reliability diagram determines what percentage of the observations lies below a certain quantile of the forecasted distribution. Ideally, j % of the observation should lie below the jth percentile of the forecasts. Additionally, confidence intervals were estimated to account for the varying amount of values per bin (Wilks, 2011). Again, forecasts were verified with 1-minute-mean SCADA data.

Also for the evaluation of probabilistic forecasts, persistence was used as a reference. Here, we generated a probabilistic persistence forecast by adding the errors of the 19 previous time steps to the forecast, as suggested by Gneiting et al. (2007).

#### 4.4.2 Unstable stratification

In Table 3 the average crps of persistence and the LF is compared for turbines T1-T7 for unstable situations for all available forecasts as well as all simultaneously available ones. Here, forecasts of turbines T1-T4 are sharper and better calibrated than persistence, while for T6 and T7 persistence outperforms the LF. When considering only simultaneously available forecasts, persistence only outperforms the LF forecast for T7. These results are in good agreement with the deterministic scores, indicating that the LF achieves better quality in unstable conditions as long as sufficient wind field data is available.

An exemplary time series of the lidar forecast for unstable stratification is shown in Figure 13. The turbine's 1-minute-mean SCADA power, the LF's mean values and persistence are plotted in blue, red and green, respectively. Each marker represents





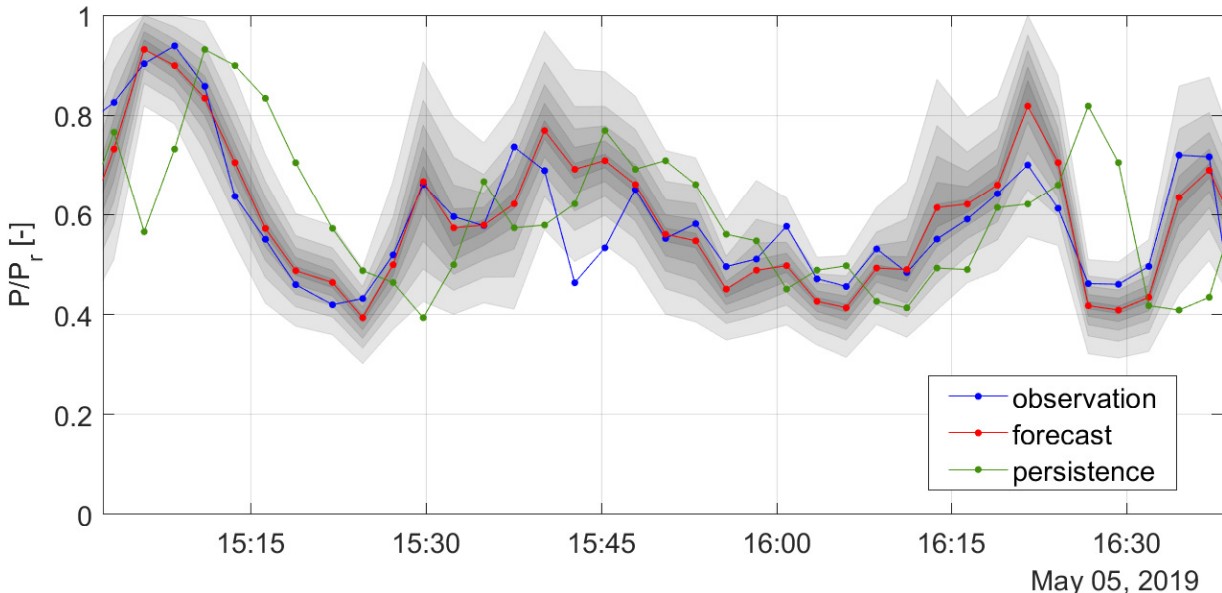

**Figure 13.** An example 1.5-hour time series of 5-minute-ahead lidar power forecast for unstable stratification at turbine T3 shown in red. Confidence intervals are visualised as shaded grey areas from $5\,\%$ to $95\,\%$ in $10\,\%$ intervals. The blue curve shows 1-minute-mean SCADA data of T3.

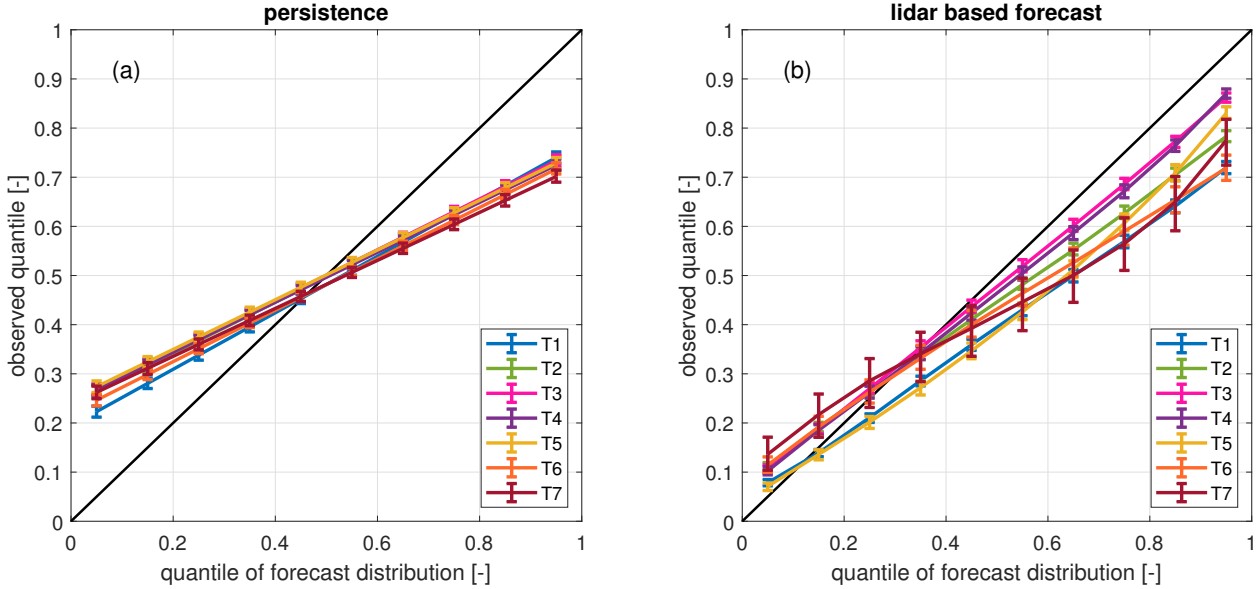

**Figure 14.** Reliability diagrams of all free flow turbines of (a) the lidar-based forecast and (b) persistence for **unstable** stratification. $95\,\%$ confidence intervals are visualised as error bars.





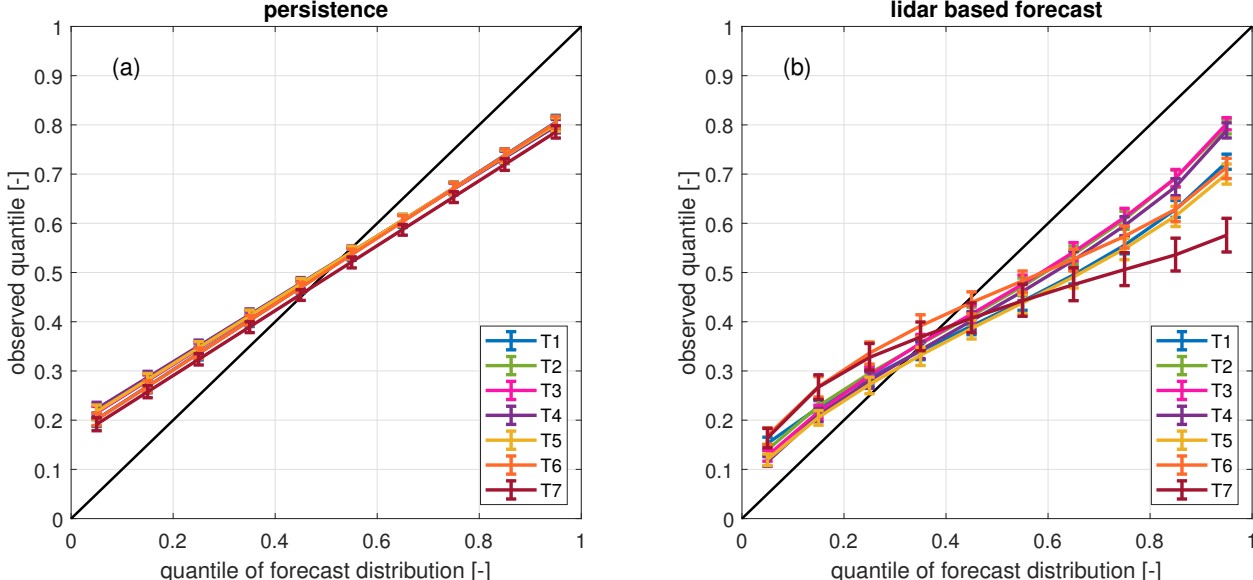

**Figure 15.** Reliability diagrams of all free flow turbines of (a) the lidar-based forecast and (b) persistence for **stable and neutral** stratification. 95 % confidence intervals are visualised as error bars.

one forecast, generated with a temporal resolution of about 2.5 minutes. Shaded grey areas around the lidar forecast's mean indicate 5 % to 95 % prediction intervals in 10 % steps. Generally, the LF is able to follow the observed power more accurately

than persistence does. Starting at 15:09 UTC a ramp event occurs, with a power drop from 92 % to 42 % within a time interval of 13.5 minutes. The LF predicts the ramp event quite accurately. Another extreme power drop of 40 % within 5 minutes can be observed at 16:21 UTC, also well captured by the lidar forecast. For both cases, persistence strongly overestimates the power. The width of the prediction intervals ranges from 18 % to 48 %. Broader intervals might be an indicator for higher uncertainties associated with the forecast. At all times except for two time steps, the intervals are able to capture the true power fluctuations.

In 26 of the 37 depicted forecasts, the observed power lies within the 25 %-75 % interval, in 5 cases within the 45 %-55 % interval.

In Figure 14 the reliability diagram of turbines T1-T7 is depicted for persistence as well as the lidar-based forecast for the unstable cases. For none of the seven turbines, persistence is well-calibrated, but shows large discrepancies to the diagonal black line, which would indicate a perfect calibration. For T3 about 27 % lie below the 5 % quantile, while only 73 % lie below

the 95 % quantile. All turbines have very similar reliability. The calibration of the LF is in general better than for persistence, especially for turbines T2 to T4. For low quantiles, for all turbines 7 % − 14 % of the LF lie below the 5 % quantile. Considering the confidence intervals assigned to those values, here the forecasts are well-calibrated. For high quantiles, the turbines show large differences in reliability. While T3 is comparatively well-calibrated with 86 % below the 95 % quantile, T7 is hardly calibrated with a value of 77 %. The generally too low values for large quantiles suggest that a higher probability needs to be

assigned to higher power values (Hamill, 1997).



**Table 3.** Number of valid forecasts $N$ and average crps for turbines T1-T7 for the probabilistic lidar-based forecast and persistence during **unstable** stratification for all available forecasts and all simultaneously available ones. The crps is given in % of the turbines' nominal power with lowest values in bold.

|  |  | T1 | T2 | T3 | T4 | T5 | T6 | T7 |
|---|---|---|---|---|---|---|---|---|
|  | N | 2510 | 2840 | 2948 | 2710 | 2048 | 640 | 297 |
| $\overline{\mathrm{crps}}$ [%] | lidar-based | **6.86** | **8.78** | **8.25** | **9.01** | **11.54** | 12.35 | 19.22 |
|  | persistence | 7.53 | 9.17 | 10.11 | 10.62 | 11.58 | **11.49** | **11.61** |
|  | N | 45 | 45 | 45 | 45 | 45 | 45 | 45 |
| $\overline{\mathrm{crps}}$ [%] | lidar-based | **10.37** | **11.51** | **10.14** | **10.94** | **12.08** | **12.09** | 18.54 |
|  | persistence | 11.41 | 12.57 | 12.93 | 14.23 | 14.03 | 12.41 | **12.60** |

**Table 4.** Number of valid forecasts $N$ and average crps for turbines T1-T7 for the probabilistic lidar-based forecast and persistence during **stable and neutral** stratification. The crps is given in % of the turbines' nominal power with lowest values in bold.

|  |  | T1 | T2 | T3 | T4 | T5 | T6 | T7 |
|---|---|---|---|---|---|---|---|---|
|  | N | 2013 | 2197 | 2209 | 2102 | 1273 | 1219 | 554 |
| $\overline{\mathrm{crps}}$ [%] | lidar-based | 10.93 | 10.46 | 10.61 | 11.14 | 15.16 | 12.78 | 22.80 |
|  | persistence | **6.13** | **6.63** | **6.60** | **6.98** | **6.83** | **6.46** | **3.89** |

### 4.4.3 Stable and neutral stratification

We compare the average crps for stable and neutral conditions of persistence and the LF in Table 4. Here, persistence is generally more accurate than the LF. Again, persistence's quality is considerably better compared to unstable situations, while that of the LF is strongly reduced. The reliability diagrams depicted in Figure 15 demonstrate that persistence is also better

calibrated in stable and neutral cases, however, still about 20 % of the forecast lie below the 5 % quantile while only about 80 % lie below the 95 % quantile. Again, all turbines show very similar results. Using the LF, especially at high quantiles, fewer observations than expected lay below the respective quantiles, indicating that higher probabilities need to be assigned to larger values (Hamill, 1997). T2, T3 and T4 are best calibrated with 79 %-80 % below the 95 % quantile. For all other turbines, in particular those positioned in areas with low lidar scan coverage, results are worse than for persistence.

## 5 Discussion

We introduced a minute-scale forecasting methodology for long-range single-Doppler lidar measurements and used it to predict the power of seven free-flow turbines of the offshore wind farm Global Tech I. The proposed model was developed as an extension of and an alternative to existing methods and is applicable to far offshore sites. Emphasis was hereby put onto the





use of single-Doppler measurements as compared to dual-Doppler set-ups. In the following, we discuss the model's ability to

skillfully predict power under different atmospheric conditions. Moreover, limitations, possibilities and necessary adjustments concerning the forecast horizon are assessed. Finally, we qualitatively analyse the forecast uncertainty.

### 5.1 Forecasting skill for different atmospheric conditions

While the LF was able to predict wind power more reliable than persistence for unstable situations, the methodology failed when applied during stable stratification. Generally, we would expect the assumptions of a homogeneous wind field and negli-

gible vertical wind speed component, that are the basis of the wind field reconstruction, to be less applicable during unstable situations when high amounts of thermal buoyancy cause strong vertical mixing (Stull, 2017). Also, the Lagrangian advection technique is expected to be more accurate for stable cases as during unstable situations vertical mixing considerably impacts the flow (Würth et al., 2018). Valldecabres et al. (2018b), for instance, found that for far ranges wind speed fluctuations are not well captured by the applied wind field reconstruction methods. This implies that especially unstable situations cannot be

predicted well. We, therefore, suppose that the low forecast skill observed during stable and neutral stratification is not related to the wind field reconstruction, but to the differences of measuring and target height. That includes especially errors in the extrapolation of wind speed to hub height. In Section 4.2 we have already shown that due to the nature of the stability corrected logarithmic wind profile a misestimation of the Obukhov length $L$ may have a strong impact on the forecast quality in stable situations. Other errors are possibly caused by an inaccurate estimation of the roughness length, a wrong estimation of the ac-

tual measuring height or a general inapplicability of the logarithmic wind profile. A more detailed analysis of errors associated with wind speed extrapolation of long-range lidar measurements by Theuer et al. (2020) supports this interpretation.

Further, the height difference causes errors in wind vector advection. We chose to propagate wind vectors to the target turbines prior to the wind speed extrapolation. Vectors were thus propagated with the lower wind speed at measuring height compared to that at hub height. This suggests wind vectors arrive at the turbines slightly delayed, with the extend of the delay

related to the measuring height. We assume this reduces the forecast skill. As the increase of wind speed with height is larger during stable stratification, we expect the effect to be more distinct for those cases. In this case study, the alternative, i. e. extrapolating wind speed before propagation, caused even larger errors compared to the ones presented in Section 4. That means, here the propagation of wind vectors associated with large errors due to wind speed extrapolation has a stronger impact on the forecast accuracy than the advection at lower heights.

For future applications, a more accurate description of the wind profile, especially in stable situations, is required to further improve the forecast skill. That also includes a more accurate estimation of stability and therefore demands reliable meteorological measurements. Additional profile information could, for example, be collected using range height indicator (RHI) lidar scans or data from a nearby met mast.

While the benchmark persistence yields good forecasts for stable and neutral situations, it has obvious shortcomings for

strongly fluctuating situations and ramp events. Its comparison to the LF model has shown the latter's ability to predict such situations better (Figure 13). Especially the probabilistic forecast has proven to be more skilful compared to persistence as it



provides better-calibrated estimations of prediction intervals. We thus consider the lidar-based forecast a valuable addition to the benchmark persistence during unstable situations.

## 5.2 Forecast horizon and scanning trajectory

In this work, we developed a 5-minute-ahead power forecast. In order for remote sensing-based forecasts to be useful for power grid balancing and electricity trading, the forecast horizon needs to be extended further (Würth et al., 2019). The accuracy of the lidar-based forecasts is expected to decrease with increasing lead times, however, Würth et al. (2018) found the accuracy of the state-of-the-art persistence to decrease faster. Lidar-based forecasts thus have the potential to bridge the gap between persistence and hour-ahead forecasts.

Small lidar systems suitable for offshore campaigns typically reach measurement distances from $8\,\text{km}$ to a maximum of $10\,\text{km}$ (Leosphere, 2018). Assuming a measuring distance of $9\,\text{km}$ from the turbine's position and not considering reductions of lead time due to time synchronisation, LFs can be used to predict a $10\,\text{m\,s}^{-1}$ mean wind speed with variations of $\pm 20\,\%$ with a lead time of 12.5 minutes, $8\,\text{m\,s}^{-1}$ mean wind can be predicted 15.6 minutes ahead. However, not only the maximal measurement distance and wind speed but also the wind farm layout, scan geometry and wind direction have a significant

impact on the lead time and quality of the forecast of individual turbines. We showed that forecasts for turbines positioned in an area not covered well by the lidar scan show low quality and fewer situations can be forecasted compared to those placed in a well-covered area. Due to the limited area, a smaller number of wind vectors can be advected to the turbine of interest, resulting in lower forecast availability and larger biases. This is confirmed by Valldecabres et al. (2018a) who showed how reduced radar availability decreases the maximal lead time, number of valid forecasts and quality of the forecast's calibration.

A large disadvantage of single-Doppler lidar data is the need to exclude a critical region with $75° < |\vartheta - \chi| < 105°$ as a consequence of the VAD fit (Section 3.1), which enhances this effect. The extent to which specific turbines are affected also depends on wind direction. Additionally, an inaccurate adjustment of the scanning trajectory to the wind direction can reduce data availability.

Furthermore, the long duration of the scans in this analysis caused the need for a time synchronisation and reduced the
achievable forecast horizon after the end time of the scan significantly (Section 3.3). Possibilities to reduce the scan time are i) an increased scanning speed, which reduces the maximum measurement distance, ii) a lower azimuthal resolution, which introduces errors to the wind field interpolation especially for far range gates, and iii) a reduced total azimuth spanned, which further reduces forecast availability and quality for some of the target turbines. To make reliable statements regarding the optimal lidar position and scanning trajectory, a more detailed analysis of forecast quality for different wind directions and
scan geometries is necessary. This should also include a study on the effect of reduced scanning time on forecast skill.

## 5.3 Uncertainty estimation and data availability

We already mentioned the errors attributed to the extrapolation of wind speeds to hub height, namely uncertainties in stability and wind profile estimation as well as an inaccurate determination of measuring height. While measuring at hub height would reduce the need for a wind speed extrapolation, it would introduce new challenges as the correction for significant stationary





and dynamic inclination of the scan plane due to the flexibility of the wind turbine tower and its dynamic excitation. In our case study, we had to correct for wind speed dependent platform inclination despite the fact that the lidar was positioned on a comparably stiff platform on a tripod foundation of the offshore turbine at GT I.

We further have to consider errors during the wind field reconstruction, including the estimation of global wind direction by means of a VAD fit, assuming a homogeneous flow and neglecting the vertical wind component. Those wind direction
errors, uncertainties in azimuth, elevation and range gate of the lidar system, as well as errors of the measured line-of-sight velocities, all contribute to the uncertainties in the estimation of the horizontal wind field. The use of dual-Doppler instead of single-Doppler data would allow for a more accurate estimation of horizontal wind speed components and would likely decrease the associated errors significantly. We further expect the propagation of wind vectors by means of their local wind speed and direction, both assumed constant along the entire trajectory, to introduce some uncertainties, enhanced by the errors
assigned to its input parameters. Another large contribution to the overall forecast error is the transformation from wind speed to power values, as uncertainties in wind speed are magnified due to the cubic nature of the power curve. As discussed earlier, we found the above-mentioned uncertainties to not only depend on the lidar set-up, but also on atmospheric condition. Detailed knowledge of the forecast uncertainty is important to be able to further assess the possibilities and limitations of the proposed method and to reduce sources of error.

The variety of uncertainties associated with the model emphasises the importance of the probabilistic approach as it allows us to account for some of them. The area of influence hereby plays a crucial part to determine the probabilistic forecast. The AoI estimated in this case study is five times smaller than the one Valldecabres et al. (2018a) defined in their work, despite applying the same methodology. We explain this by the many factors influencing the crps and consequently AoI, i. e. the lidar wind field, the SCADA time series and the number of wind vectors available to be propagated. The difference in AoI suggests
that it needs to be determined individually for each data set.

As already mentioned, the VAD fit caused the estimated wind direction to be constant across azimuth angles and only vary with range gates. This likely had an impact on the individual wind vectors reaching the area of influence. The uniform wind direction across range gates restricted the area from which vectors could be propagated to the target turbines. We assume this led to a misestimation, most likely an underestimation, of the spread of the observed wind speeds. Consequently, it is
anticipated that the spread of the forecasted wind power distribution is too small. When using dual-Doppler measurements, wind directions could be determined individually for each measurement point and the forecast's distribution represented more accurately.

Another limitation of the LF is its need for high data availability. Lidars send out laser pulses and use the backscattered signal to estimate wind speed. If not enough or too many aerosols are in the air, the signal becomes noisy (Newsom, 2012).
That means e. g. during rain and fog, no accurate lidar measurements will be available and no forecast can be generated. One solution might be the development of a hybrid method that does not solely depend on the availability of lidar measurements.



## 6 Conclusions

We developed a methodology to forecast wind power of individual wind turbines on very short time horizons based on single-Doppler long-range lidar scans, as a feasible alternative to existing remote-sensing based forecasts that is applicable to far offshore sites. The work is based on a probabilistic forecasting model developed for dual-Doppler radar measurements. It was extended to include a dynamic filtering approach, a time synchronisation of the lidar scans and an extrapolation of wind speeds to hub height. The model was tested in a case study at the offshore wind farm Global Tech I. Here, we predicted wind power of seven free flow wind turbines with a 5-minute horizon. The lidar-based forecast was able to predict wind turbine power skillfully compared to the benchmark persistence during unstable atmospheric conditions, as long as sufficient wind field information was available in the region from which the wind vectors were propagated to the turbine of interest. During stable and neutral conditions the forecast quality was reduced. We mainly attribute this to higher uncertainties in the wind speed extrapolation to hub height during stable conditions, as a consequence of the nature of the stability corrected logarithmic wind profile. To outperform persistence for stable situations a more accurate description of the wind profile, e. g. using reliable meteorological information, is required.

Future work aims to include the modelling of wake effects in the forecast, allowing to forecast power not only for free-flow turbines.

*Data availability.* Lidar data could be made available on request. GT I SCADA data is confidential and therefore not available to the public.

*Author contributions.* Frauke Theuer performed the main research and wrote the paper. Marijn Floris van Dooren contributed to the scientific discussion, the outline and review of the manuscript. Lueder von Bremen and Martin Kühn supervised the research, contributed to the scientific discussion, the research concept and the outline and thorough review of the manuscript.

*Competing interests.* The authors declare no conflict of interests.

*Acknowledgements.* The lidar measurements and parts of the work were performed within the research projects OWP Control (Ref. Nr. 0324131A) and WIMS-Cluster (Ref. Nr. 0324005) funded by the German Federal Ministry for Economic Affairs and Energy on the basis of a decision by the German Bundestag. We acknowledge the wind farm operator Global Tech I Offshore Wind GmbH for providing SCADA data and thank them for supporting our work. We thank the Met Office for making the OSTIA data set available. We further acknowledge the German Federal Environmental Foundation (DBU) as this project receives funding within the scope of their PhD scholarship program.

We thank Jörge Schneemann and Stephan Voß for conducting the measurement campaign and supporting our lidar data analysis and Andreas Rott for his help characterising the lidar misalignment.



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
