# Peer review of "Minute-scale power forecast of offshore wind turbines using single-Doppler long-range lidar measurements"

_Wind Energy Science, 2020_

## Referee Comment (RC1) · Anonymous Referee #1 · 5 Jun 2020

General Comment:

This is a very well written and thorough study looking at using a single doppler lidar for minute-scale wind/power forecasts. The authors took a very systematic approach to the study, leveraging previous work/studies. They especially did a good job discussing and considering the possible causes of their results, especially when results were not favorable (stable conditions). I only have a few minor questions/ comments.

Specific Comments:

1) Lidar is still reliant on the presence of aerosols. While sea salt is often present in marine environments, the author should acknowledge that issue (line 45/46)

[Figure]

2) Line 253: were these algorithms run in real time? If so, what were you interpolating the SST measurements to past noon? Was it interpolated using the trend in SST? If this algorithm was not run in real time, will a buoy need to be deployed in addition to the lidar to provide SST measurements when applied in real time? Please add additional detail.

Technical corrections:

1) I would advise against using words like "very" as it has no concrete meaning. Example in line 32

---

## Referee Comment (RC2) · Anonymous Referee #2 · 8 Jun 2020

**Theuer et al., 2020 – Review**

General comments

The paper represents a sound scientific manuscript. The methodology is explained with a great deal of details and state-of-the-art LiDAR processing techniques are used. The specific use of relevant bibliographic sources is also commendable. Probably some discussions about the data availability could have been shortened for the sake of brevity.  The only major issue that needs to be addressed before the publication in my opinion is the presence of wake affecting the turbines under investigation for wind direction close to 350° and 130°.

Specific comments

Lines 27-28: the statement "ARIMA models additionally consider past forecasting errors" is questionable. The AM part of the ARMA model already considers the error. The ARIMA is a generalization of an ARMA which has on the right-hand side of the model a derivative of the variable. Also, in the reference (Kavasseri and Seetharaman, 2009) I could not find such claim.

Line 68: it would be useful a summary of the content of each section at the end of the introduction.

Line 121: the original VAD technique (Browning and Wexler 1968) was conceived for full 360 PPI scans. Is there any reference for the estimation of the loss of accuracy due to the utilization of partial sector of 150°?

Line 148: could the author add a comment or a reference clarifying why the backward propagation needs to be avoided?

Line 151: it is correctly stated that the difference between the synchronization time and the forecast time $(t_{n+1} - T_r + k)$ has to be minimized in order to have the observation as close as possible to the prediction. This "optimization" is however done with the constraint that no backward propagation is used in the reconstruction of the synchronized velocity field. Is this constraint applied to the whole domain? Because from Figure 3d it seems like just a small portion of the actual field is then used. It would be useful to further clarify the choice of the synchronization time.

Eq 8: this expression is not present in Dyer 1974. Please find another refence and also address the applicability of such relationship in the offshore context.

Lines 240-241: the considered wind sector (130°-350°) include wind direction where wake affect the turbines which would invalidate the use of the far wind field for power prediction. Actually, the *mae* is higher for wind directions close to the boundaries of the interval (Fig. 9b), i.e. where wake effects are expected, and for stable ABL, when wake diffusion is hampered.

Technical comments

Figure 4: $k$ is not defined before the introduction of this figure. Add explanation in caption.

Figure 9: *mae* is not defined before the introduction of this figure. Add explanation in caption.

Based on my experience with WindCube 200S, your LiDAR seems to have a quite long rest time (it should be negligible). You may want to check your set up or contact Leosphere.

---

## Referee Comment (RC3) · Anonymous Referee #3 · 15 Jun 2020

**Minute-scale power forecast of offshore wind turbines using single-Doppler long-range lidar measurements**

Frauke Theuer, Marijn Floris van Dooren, Lueder von Bremen, and Martin Kühn

**Review**

**General Comment**

The manuscript presents a methodology for short time (5-10min) wind power forecasts based on scanning lidar measurements. The methodology is tested with a unique dataset measured at the offshore wind farm Global Tech I. Overall, the paper follows a clear structure, and details are very well described. The authors make good use of existing methods such as lidar processing techniques. Figures are well prepared (despite some minor comments below). The results are present well and are discussed in great detail. Points that should be addressed prior publication are:

1. How large is the influence of the wind farm blockage effect on your forecast? Please try to estimate the influence and discuss how it could be integrated into your forecast methodology.

2. You point out that the wind profile estimation is one major factor influencing the quality of your forecast. In the paragraph starting in line 460 you state that additional wind profile information is needed. Please comment on why you did/could not use the wind speed measurements from the wind turbine nacelle? Having the lidar wind speed measurements at ~30 m and the wind speed at hub height should result in a fairly good estimation of the profile.

**Specific Comments**
1. Page 3, line 80: "TP" has already been introduced
2. Page3, line 86: Consider adding "($\phi = 0°$)" after "horizontal PPI scan" to make the difference to the high elevation scan clearer.
3. Figure 1b: specify the origin of the coordinate system
4. Line 106: replace "causes" with "factors"
5. Line 107: missing space before "Typically"
6. Line 112: How do you justify that all measurements with CNR between -26.5 dB and -5 dB can be considered as valid?
7. Line 121: "15" out of how many?
8. Line 145 -148: Consider moving this information to the figure caption.
9. Line 222: Replace "GTI" with "Global Tech I" in the heading.
10. Section 4 and throughout the entire manuscript: Please revise the use tenses. Many part of the manuscript are written in the past tense, which is not typical for scientific publications.
11. Line 246: What is the maximal number of wind vector retrievals?
12. Figure 6: Please improve the resolution of the figure.
13. Line 256: i. e.
14. Line 267: Can you please explain where the large range of 53 m results from.

15. Line 295: Please add the percentage of available data instead of the actual numbers only.
16. Line 299: Replace "worse" with "decreased"
17. Figure 7: Figure is showing availability over distance, caption refers to "range gate". Please make this consistent.
18. Figure 9: Abbreviation "mae" not introduced
19. Figure 9: The lines are difficult to differentiate. Please try out using different colors and/or line types.
20. Line 318: missing space after "15"
21. Figure 12b: What is the reason for the plateaus in the data? They do not occur in Figure 11b.
22. Figure 15: Lines are again difficult to differentiate.
23. Line 446: measuring "at" target height?

---

## Author Comment (AC1) · 13 Jul 2020

**Minute-scale power forecast of offshore wind turbines using single-Doppler long-range lidar measurements**

**Frauke Theuer, Marijn Floris van Dooren, Lueder von Bremen, and Martin Kühn**

We thank the three referees for the time and effort they put into reviewing our work and appreciate their feedback and comments. Find below our answers to the comments and attached a pdf-file highlighting all the changes made to the manuscript using latexdiff.

**Anonymous Referee #1**

*Specific Comments:*

*1) Lidar is still reliant on the presence of aerosols. While sea salt is often present in marine environments, the author should acknowledge that issue (line 45/46)*

We acknowledge the mentioned limitations of lidars in Section 5.3, Line 328 ff..

*2) Line 253: were these algorithms run in real time? If so, what were you interpolating the SST measurements to past noon? Was it interpolated using the trend in SST? If this algorithm was not run in real time, will a buoy need to be deployed in addition to the lidar to provide SST measurements when applied in real time? Please add additional detail.*

None of the algorithms were run in real time. However, the lidar-based forecasting methodology was in general developed in a way that makes a real-time application possible. That means, it only uses past data, i. e. data that is available at the initialisation of the forecast, to generate the forecast itself. The only exception is here, indeed, the use of the OSTIA SST data. As there were no on-site water temperature measurements during the analysed time period available, we utilised the OSTIA SST data, as an estimation of stability is crucial to the height extrapolation. In our work, the OSTIA SST data was linearly interpolated, utilising both past and future values. The idea was to "mimic" a buoy measurement by doing so. That means after once having reconstructed the water temperature time series from OSTIA SST data, we utilised - just as for pressure, air temperature and humidity measurements - only "past data", i. e. time steps previous to the initialisation of the forecast. An analysis of Schneemann et al. (2020) has revealed that the agreement between the interpolated OSTIA SST and buoy measurements is good.

If one wanted to use the OSTIA SST for real-time applications, the interpolation needed to be adjusted in a way that it only utilises past data as Anonymous Referee #1 suggested. This could possibly be done by utilising the observed trend or adopting the previous value. A comparison with buoy measurements has revealed, that both alternatives reduce the water temperature accuracy significantly. Consequently, also the stability estimation and wind speed extrapolation are expected to be subject to larger errors.

To summarise, accurate stability information is important to perform an accurate height extrapolation (compare also Section 5.1, Line 460 ff.). We thus suggest to use buoy measurements for future applications instead of relying on OSTIA SST data. Also accurate and undisturbed air temperature measurements at at least two heights might be a good alternative to determine atmospheric stability. Both approaches would incur additional equipment and operational costs.

Following the reviewer's comments, we added some more detail to Section 4.1, explaining the use of OSTIA SST data more precisely:

Interpolations were performed utilising both past and future values with respect to the initialisation time. The interpolated SST data is in this context understood as an artificial buoy measurement.

We also added an additional paragraph to the discussion in Section 5.1, Line 462.

We thus suggest to use buoy measurements for future applications instead of relying on OSTIA SST data. Also accurate and undisturbed air temperature measurements at at least two heights might be a good alternative to determine atmospheric stability. Both approaches would incur additional equipment and operational costs.

*Technical corrections:*

*1) I would advise against using words like "very" as it has no concrete meaning. Example in line 32*

We have omitted the word "very" in several cases. However, we kept the expression "very short-term forecast" (for example in Line 32) as this is a commonly used expression to describe minute-scale forecasts (compare for instance Dowell and Pinson, 2016; Valddecabres et al., 2018a; Valldecabres at al., 2018b; Sweeney et al., 2019).

**Anonymous Referee #2**

*Specific comments:*

*Lines 27-28: the statement "ARIMA models additionally consider past forecasting errors" is questionable. The AM part of the ARMA model already considers the error. The ARIMA is a generalization of an ARMA which has on the right-hand side of the model a derivative of the variable. Also, in the reference (Kavasseri and Seetharaman, 2009) I could not find such claim.*

We have corrected the error and adjusted the reference:

Other statistical models such as ARMA (autoregressive moving average) take a higher number of past values and also past forecasting errors into account (Torres et al., 2005). ARIMA (autoregressive integrated moving average) models additionally difference the time series to achieve stationarity (Grigonytė, 2016).

*Line 68: it would be useful a summary of the content of each section at the end of the introduction.*

We added a short summary of each section at the end of the introduction in Section 1, Line 67:

The manuscript is structured as follows: Section 2 describes lidar scans used for minute-scale forecasting. In Section 3 the forecasting methodology is developed. Section 4 provides an overview of the case study analysed here, evaluates the proposed methodology and presents the results of probabilistic as well as deterministic power forecasts. In Section 5 we discuss possible sources of uncertainty and the impact of atmospheric stability and the measurement set-up on the results before the conclusions (Section 6) are drawn.

*Line 121: the original VAD technique (Browning and Wexler 1968) was conceived for full 360 PPI scans. Is there any reference for the estimation of the loss of accuracy due to the utilization of partial sector of 150°?*

When applying a VAD-like reconstruction methodology to PPI scans it is common to use sector scans (Valldecabres et al., 2018; Beck and Kühn, 2019; Schneemann et al. 2020) instead of a full 360° scan. Simon (2015) analysed the optimum sector

size for PPI lidar scans, investigating sector sizes ranging from $4°$ to $60°$. He found the results not to vary significantly when comparing $30°$ and $60°$ scans. We are not aware of any other studies that systematically analyse the impact of sector size on the wind speed accuracy.

Our analysis of PPI scans has shown that the applied methodology is very robust, we thus don't expect significant errors caused by sector size. In order to minimise errors associated with the VAD fit we neglect range gates with low data availability. We also performed a "mirroring" of the data points (based on the symmetry of the sine curve) to enhance the fit's robustness.

We adjusted Line 119f. in Section 3.1 to:

After filtering, the global wind direction was determined by performing a VAD-like fit individually for each range gate in a certain scan.

***Line 148: could the author add a comment or a reference clarifying why the backward propagation needs to be avoided?***

The backward propagation that we mention in Line 148 refers to that of a future scan measured within the time period $[t_{n+1}, t_{n+1} + T_\vartheta]$. As this scan was not yet measured at the initialisation time of the forecast at $t_{n+1} - T_r$, it cannot be utilised for the time synchronisation (Figure 4). However, a backward propagation of the current scan, measured within the period $[t_n, t_n + T_\vartheta]$, is used for the time synchronisation. To make this distinction between backward propagation of future and current scans more clear, we adjusted Line 148 as follows:

For the purpose of forecasting, $t_{syn}$ should be chosen to stay within the region of the weighting function that puts no weight to backwards propagated scans, that were measured after the forecast's initialisation time, thus $t_{syn} \in [t_n, t_n + a\,\Delta T]$.

***Line 151: it is correctly stated that the difference between the synchronization time and the forecast time $t_{n+1} - T_r + k$ has to be minimized in order to have the observation as close as possible to the prediction. This "optimization" is however done with the constraint that no backward propagation is used in the reconstruction of the synchronized velocity field. Is this constraint applied to the whole domain? Because from Figure 3d it seems like just a small portion of the actual field is then used. It would be useful to further clarify the choice of the synchronization time.***

We hope our answer to the previous comment provides clarification regarding the backward propagation. In our work we utilised a weighting function introduced by Beck and Kühn (2019), that puts no weight to the backward propagated scan during the first fifth of the total scanning time. As we cannot use a future scan to perform a backward propagation (see previous comment), our synchronisation time needs to lie within that time period (referred to as $[t_n, t_n + a\,\Delta T]$ in Section 3.2).

Figure 3d visualises an example for wind vectors contributing to a forecast at turbine T3, thus advecting to the target turbine within the specified time frame. Which portion of the wind field is used depends on wind speed, wind direction, lead time, the area of influence and the position of the target turbine. We agree that for future applications it would make sense to adjust the measurement set-up to better utilise the scanned area (Section 5.2).

***Eq 8: this expression is not present in Dyer 1974. Please find another reference and also address the applicability of such relationship in the offshore context.***

We have changed the corresponding reference to Paulson, 1970 and Holtslag and De Bruin, 1988.

The stability correction term we provide here is, in combination with the Charnock relation (Equation 4), widely used in an offshore context, for instance refer to Sanz Rodrigo et al. (2015), Pena and Gryning (2008) and Lange et al. (2004). Deviations

between profiles and observation can mainly be observed during stable atmospheric conditions. A detailed analysis of this is considered not to be within the scope of this paper but was investigated in more detail by Theuer et al. (2020) as also stated in Section 5.1.

120

*Lines 240-241: the considered wind sector (130°-350°) include wind direction where wake affect the turbines which would invalidate the use of the far wind field for power prediction. Actually, the mae is higher for wind directions close to the boundaries of the interval (Fig. 9b), i.e. where wake effects are expected, and for stable ABL, when wake diffusion is hampered.*

125 We agree that for wind sectors close to 130° and 350° wind turbines might be affected by wakes. In order to account for possible wakes our algorithm neglects wind vectors that i) originate from within the wind farm and ii) have already passed through the area of influence of another turbine. This is based on the assumption that turbines are not necessarily placed within the wake of another turbine for the mentioned wind directions. It is thus assumed that vectors, which are able to travel to the target turbine unhindered, can contribute to the wind speed forecast.

130 As the area of influence is rather small in this case study ($R_\mathrm{AoI} = 23.2\,\mathrm{m}$), many wind vectors will contribute to the forecast despite this filter and whenever the threshold of $N = 20$ is reached the forecast will be considered valid. We are aware that this representation of wakes is a very simplified one and that more research is required in this field, especially when aiming to forecast power of turbines placed within the wind farm. However, this is not within the scope of this paper.

In the context of Anonymous Referee #2's comment it is further important to consider that wind directions are dependent on
135 range gate. That means, the wind directions used to advect wind vectors and thus the wind direction of wind vectors contributing to the forecast, might differ from the mean wind direction that is used for filtering.

To summarise, we agree that we need to mention the possible implication of wakes, when explaining Figure 9 and also in the discussion. We have thus added the following explanations to Section 4.2, Line 308 and 5.3, Line 510 respectively:
140 For wind directions larger than 310° an increasing mae and a decreasing number of valid forecasts can be observed for turbines T2-T7. This is likely related to the interference with wakes. Due to the wind farm layout some vectors were advected through the wind farm area before reaching the target turbine. Even though wind vectors blocked by other turbines were not considered here, this simple advection technique cannot represent the more complex flow within the wind farm. A similar problem occurs for wind directions smaller than 150°, in this case mainly affecting T1-T5.

145

Especially in situations where wind vectors were partially propagated through the wind farm area and thus turbines might have been affected by wakes, large errors were observed.

As also the selection of wind vectors was not stated clearly up to now we added it to the manuscript in Section 3.4, Line 163.
150 Hereby, wind vectors originating inside of the wind farm area were neglected. Further, we considered vectors to be able to only contribute to one turbine, i. e. the first turbine they reached.

At this point we would also like to highlight that the distinct quality differences of the LF for varying atmospheric conditions are not an artefact of wakes at the boundaries of the wind direction intervals as suggested by Anonymous Referee #2. These
155 situations only have a small impact on the overall scores presented in Section 4.3 and 4.4. Neglecting them would still result in

significantly larger errors during stable and neutral situations as compared to unstable ones. That means, while it is important to acknowledge and explain the difficulty to forecast such situations, this does not put any of the conclusions drawn in this work into question.

160 *Technical comments*

*Figure 4: k is not defined before the introduction of this figure. Add explanation in caption.*

$k$ is firstly introduced in Figure 2, however, we added an explanation to the caption of Figure 4 and the description of the methdology in Section 3.

165 *Figure 9: mae is not defined before the introduction of this figure. Add explanation in caption.*

We added an explanation in the caption of Figure 9 and Section 4.2.

*Based on my experience with WindCube 200S, your LiDAR seems to have a quite long rest time (it should be negligible). You may want to check your set up or contact Leosphere.*

170 Thanks for pointing this out. To our knowledge the reset time of a few seconds, required for the scanning head to return to its initial position, is normal and is predominantly limited by the maximum angular rate of the scanner.

Anonymous Referee #3

*General Comments*

175 *1. How large is the influence of the wind farm blockage effect on your forecast? Please try to estimate the influence and discuss how it could be integrated into your forecast methodology.*

This is an interesting question and the topic of wind farm blockage is increasingly gaining attention. Several studies have suggested the existence of a wind farm blockage effect and indicated that, even though the effect is small compared to the influence of wake effects, it might have an impact on the power production of a wind farm (Bleeg et al.,2018; Segalini and

180 Dahlberg 2019). Not considering the effect would thus lead to an overestimation of power. At this stage of research no models have been developed to describe the blockage effect in more detail. Many factors likely have an impact as for instance wind farm layout, wind direction, operating conditions of the wind turbine/wind farm and atmospheric stability. It is thus hard to estimate the influence the blockage effect has on our forecast and also an integration of the effect into the forecast is at this stage not possible. One option to account for blockage would be to develop and apply an empirical model specifically designed

185 for the wind farm of interest. However, it would require a large amount of data to do so.

We added a sentence on wind farm blockage into our discussion in Section 5.3, Line 510.:

Moreover, recent studies suggest the existence of a wind farm blockage effect (Bleeg et al., 2018), which might cause wind vectors to slow down when approaching the wind farm.

190 *2. You point out that the wind profile estimation is one major factor influencing the quality of your forecast. In the paragraph starting in line 460 you state that additional wind profile information is needed. Please comment on why you did/could not use the wind speed measurements from the wind turbine nacelle? Having the lidar wind speed measurements at 30 m and the wind speed at hub height should result in a fairly good estimation of the profile.*

We agree, there are a numerous possibilities how to determine the profile information that is then used for the extrapolation of
wind speed to hub height in Section 3.4. One of them is the use of simultaneous wind speed measurements at two (or more)
different heights at the same location, that can then be fitted to a logarithmic (or any other) profile. One particular difficulty
is given by the fact that the different measurements by lidar and nacelle anemometry as well as the lidar measurements at
different range gates do not satisfy these conditions. In our work we decided to adopt a physical approach, i. e. an estimation
of stability based on meteorological measurements. Here, we used water and air temperature measurements to do so, however,
another option would be to utilise temperature measurements at two different heights. A more detailed analysis of the effect
of different profile estimation approaches on the accuracy of the forecast would have gone beyond the scope of this paper. We
are, however, aware of the problem and currently investigating it in more detail. First results on the accuracy of a logarithmic
extrapolation of wind speed measured by scanning lidars were published by Theuer et al. (2020). More detailed research on the
implications for forecasting, also investigating the use of SCADA data for vertical profile estimation, is currently conducted by
the authors.

*Specific Comments*
*1. Page 3, line 80: "TP" has already been introduced*
We have adjusted this.

*2. Page3, line 86: Consider adding "($\Phi =°$)" after "horizontal PPI scan" to make the difference to the high elevation scan clearer.*
We have adjusted this.

*3. Figure 1b: specify the origin of the coordinate system*
We have added an explanation on the origin of the coordinate system in the caption of Figure 1b.
The lidar is positioned on the transition piece (TP) of turbine T2 marked in red and defined as the origin of the coordinate system.

*4. Line 106: replace "causes" with "factors"*
We have adjusted this.

*5. Line 107: missing space before "Typically"*
We have adjusted this.

*6. Line 112: How do you justify that all measurements with CNR between -26.5 dB and -5 dB can be considered as valid?*
This choice is based on similar CNR-thresholds suggested in literature (Valldecabres et al., 2018b; Würth et al., 2018; Karagali et al., 2018). We tested different commonly used thresholds and chose one, that was in good agreement with our measurement data.

We added a sentence explaining this in Section 3.1, Line 113:

Our choice of CNR-thresholds is hereby based on similar ones suggested in literature (Valldecabres et al., 2018b; Würth et al., 2018).

*7. Line 121: "15" out of how many?*

The maximal number of values is defined by the azimuth angles measured, here 75 values.

*8. Line 145 -148: Consider moving this information to the figure caption.*

In our opinion, Lines 145–148 increase readability and clarity of Section 3.2, we thus chose to keep the description of Figure 4 both in the text as well as in the figure caption.

*9. Line 222: Replace "GTI" with "Global Tech I" in the heading.*

We have adjusted this.

*10. Section 4 and throughout the entire manuscript: Please revise the use tenses. Many part of the manuscript are written in the past tense, which is not typical for scientific publications.*

We appreciate this observation. To our knowledge the use of past tense is suggested when describing results as well as methods (for instance refer to https://www.nature.com/scitable/topicpage/effective-writing-13815989/ or https://wordvice.com/which-verb-tense-to We have checked the consistent use of tenses in our manuscript and adjusted it in some cases.

*11. Line 246: What is the maximal number of wind vector retrievals?*

All vectors fulfilling the criteria will contribute to the wind speed forecast, i. e. no upper threshold was defined. In our case study the maximal number of contributing wind vectors observed was 179.

*12. Figure 6: Please improve the resolution of the figure.*

We have adjusted this.

*13. Line 256: i. e.*

We have adjusted this.

*14. Line 267: Can you please explain where the large range of 53 m results from.*

Varying measuring heights are mainly caused by the tilt of the lidar device (Line 255-265). The tilt was characterised using available sea-surface-levelling scans and found to depend on the thrust acting onto turbine T7. It is thus variable with time and was estimated individually for each scan by means of a correction function. Depending on the power production at T58, the wind direction and the origin of the contributing wind vectors measuring heights are therefore varying quite significantly during the analysed time period.

Height variations are further introduced by the curvature of the Earth, which increases measuring height especially in further distances. However, this impact is rather small compared to the lidar's tilt. For instance in 8 km distance, the difference to the TP height yields approximately 5 m.

270    We have also noticed that we put a wrong value here, minimal heights were observed to be 20 m not 12 m. We corrected that in the manuscript.

**15. Line 295: Please add the percentage of available data instead of the actual numbers only.**
We have adjusted this.

275

**16. Line 299: Replace "worse" with "decreased"**
We have adjusted this.

**17. Figure 7: Figure is showing availability over distance, caption refers to "range gate". Please make this consistent.**
280    The caption refers to the range gate defined as $r$ in Section 3.1. In accordance with that, Figure 7 shows the availability over $r$, i. e. the range gate. Also in the text we always refer to "range gate" never to "distance". In our opinion, we use the term "range gate" consistently.

**18. Figure 9: Abbreviation "mae" not introduced**
285    We have adjusted this.

**19. Figure 9: The lines are difficult to differentiate. Please try out using different colors and/or line types.**
We adjusted the line colors in Figure 9, Figure 14 and Figure 15 to be more easily distinguishable.

290    **20. Line 318: missing space after "15"**
We have adjusted this.

**21. Figure 12b: What is the reason for the plateaus in the data? They do not occur in Figure 11b.**
The plateaus in the data are an artefact of the transformation from wind speed to power values (Section 3.5). Wind speed val-
295    ues are sorted into intervals of $0.5\,\mathrm{m\,s^{-1}}$ width. For each bin a probability distribution of the corresponding power values was constructed. In cases where (almost) all wind vectors are placed within the same wind speed bin the power values derived from that will resemble the probability distribution of the power curve. The Figure shows the deterministic forecast, i. e. the mean of the probabilistic forecast, and thus plateaus occur at positions that correspond to the mean of the power curve's probability distribution. As in unstable situations the wind vectors are more likely spread over several wind speed intervals, these plateaus
300    are less distinct in Figure 11b.

We added a brief explanation of this effect in Section 4.3.3, Line 371.
The plateaus, that can be observed in Figure 12 (b), are an artefact of the transformation from wind speed to power. In cases where most wind vectors are placed within the same wind speed bin, the forecasted power will be close to the average power
305    value of the corresponding wind speed interval (Figure 5).

**22. Figure 15: Lines are again difficult to differentiate.**

See response to comment #19.

**23. Line 446: measuring "at" target height?**

In Section 51, Line 446 we refer to the differences between measuring height and target height (hub height). We adjusted the sentence for more clarity:

We, therefore, suppose that the low forecast skill observed during stable and neutral stratification is not related to the wind field reconstruction, but to the differences of measuring height and target height.

**Further changes in addition to the reviewers's comments**

After consultation with Valldecabres et al. (2018b) and Valldecabres et al. (2018a) we further specified statements made in Section 5.1, Line 443f. and Section 5.2, Line 482f..

Valldecabres et al. (2018b), for instance, found that for far ranges the applied wind field reconstruction methods, more specifically the higher arc length, could act as a low pass filter and consequently smooth out the wind speed fluctuations.

This is confirmed by Valldecabres et al. (2018a) who showed how reduced radar availability reduces the wind speeds that could be forecasted, the number of valid forecasts and quality of the forecast's calibration.

We further clarified the introduction and interpretation of confidence intervals in Section 4.4.1, Line 391f. and Section 4.4.2, Line 416f. respectively.

Additionally, confidence intervals were estimated by means of a resampling technique to account for the varying amount of values per bin and the varying number of valid forecasts for the different turbines (Hamill, 1997; Wilks, 2011).

Taking also the narrow confidence intervals assigned to those values into account, here the forecasts are comparatively well-calibrated.

Moreover, we noticed an error in Figure 13, which we corrected, and also added a description of persistence in the figure caption.

[revised manuscript text omitted]

---

## Author Response (AR2)

**Minute-scale power forecast of offshore wind turbines using single-Doppler long-range lidar measurements**

**Frauke Theuer, Marijn Floris van Dooren, Lueder von Bremen, and Martin Kühn**

5

Dear Mr Mann,

thank you for your comments. Please find our answers below.

1) The black vector (arrow) we show in Figure 3 (d) indicates the mean wind direction within the scan. What we refer to as *point cloud of wind vectors* is the (in this cased) blue area shown. This might be a little hard to distinguish, but it consists of a number of individual points, i. e. wind vectors, positioned on the Cartesian grid described in Section 3.1. All those vectors will reach the red turbine in 5 minutes±30 seconds.

2) We changed the caption of Figure 11 and 12 accordingly.